# Molecular basis of RNA-binding and autoregulation by the cancer-associated splicing factor RBM39

Sébastien Campagne [1,2,6] ✉, Daniel Jutzi [3,6], Florian Malard[1,2], Maja Matoga [1], Ksenija Romane [1], Miki Feldmuller[1], Martino Colombo[4,5], Marc-David Ruepp [3] ✉ & Frédéric H-T. Allain [1] ✉

Pharmacologic depletion of RNA-binding motif 39 (RBM39) using aryl sulfonamides represents a promising anti-cancer therapy but requires high levels of the adaptor protein DCAF15. Consequently, novel approaches to deplete RBM39 in an DCAF15-independent manner are required. Here, we uncover that RBM39 autoregulates via the inclusion of a poison exon into its own pre-mRNA and identify the *cis*-acting elements that govern this regulation. We also determine the NMR solution structures of RBM39's tandem RNA recognition motifs (RRM1 and RRM2) bound to their respective RNA targets, revealing how RRM1 recognises RNA stem loops whereas RRM2 binds specifically to single-stranded N(G/U)NUUG. Our results support a model where RRM2 selects the 3'-splice site of a poison exon and the RRM3 and RS domain stabilise the U2 snRNP at the branchpoint. Our work provides molecular insights into RBM39-dependent 3'-splice site selection and constitutes a solid basis to design alternative anti-cancer therapies.

Alternative pre-mRNA splicing is a crucial step of gene expression that not only increases the coding capacity of the genome but also regulates the transcriptional output of genes. Consequently, non-physiological splicing patterns contribute significantly to diseases, in particular to cancer malignancy[1,2]. Understanding how aberrant splicing patterns drive tumorigenesis and the survival of cancer cells is crucial for the development of novel therapeutic approaches[3,4].

The RNA-binding protein called RNA Binding Motif 39 (RBM39) is overexpressed in several types of cancer[5–7] and is essential for the survival of many cancer cells including Acute Myeloid Leukaemia (AML) cells[8]. In AML, RBM39 sustains a network of RNA-binding proteins by splicing of their pre-mRNAs and ensures the correct processing of pre-mRNAs encoding Homeobox protein A9 targets[9–11]. As a consequence, anticancer drugs coined the aryl sulfonamides, that specifically induce the targeted degradation of RBM39, trigger the

death of cancer cell lines derived from hematopoietic and myeloid lineages[12–14] but also cancer stem cells[15] and recently, showed exceptional responses in the treatment of high-risk neuroblastoma models[16,17]. The small molecule acts as a molecular glue between RBM39 and the DCAF15-CRL4 E3 ubiquitin ligase to induce proteasome-mediated degradation of RBM39 which in turn leads to cancer cell death[18–20]. Furthermore, the targeted degradation of RBM39 generates bona fide neoantigens and augments checkpoint immunotherapy[21]. However, a major caveat of this innovative approach is the dependence on DCAF15, the adaptor protein acting at the interface between the aryl sulfonamide and the E3 ubiquitin ligase activity[16,22]. Hence, the efficiency of this treatment correlates with DCAF15 expression levels. Accordingly, the control of RBM39 homeostasis is important for cancer cell survival, and therefore the identification of alternative approaches to lower the intracellular

[1]ETH Zurich, Department of Biology, Institute of Biochemistry, 8093 Zurich, Switzerland. [2]University of Bordeaux, Inserm U1212, CNRS UMR5320, ARNA Laboratory, 33077 Bordeaux, France. [3]United Kingdom Dementia Research Institute Centre, Institute of Psychiatry, Psychology and Neuroscience, King's College London, Maurice Wohl Clinical Neuroscience Institute, London SE5 9NU, UK. [4]University of Bern, Department of Chemistry and Biochemistry, 3012 Bern, Switzerland. [5]Celgene Institute of Translational Research in Europe (CITRE), Bristol Myers Squibb, 41092 Seville, Spain. [6]These authors contributed equally: Sébastien Campagne, Daniel Jutzi. ✉e-mail: sebastien.campagne@inserm.fr; marc-david.ruepp@kcl.ac.uk; allain@bc.biol.ethz.ch

concentration of RBM39 independently of DCAF15 could be beneficial for cancer treatment.

RBM39 is a ubiquitously expressed SR-like protein and is homologous to U2AF2[23] (Supplementary Fig. 1). The protein consists of an N-terminal RS domain followed by two putative RNA Recognition Motifs (RRM1 and RRM2). A third RRM (RRM3), sometimes referred to as the U2AF2-homology motif (UHM), has been evolutionarily repurposed to mediate protein–protein interactions with U2AF2 or the U2 snRNP component SF3B1[24,25]. These contacts explain the presence of RBM39 in early spliceosomal complexes A, B and E[26–28] and indicate how RBM39 could modulate gene expression during pre-mRNA splicing[29–31]. In contrast to U2AF2, which specifically binds to polypyrimidine tracts[31], cross-linking and immunoprecipitation (CLIP) experiments suggest that RBM39 has a different RNA-binding selectivity, although the proposed sequence motifs show little overlap[8,32]. Hence, the RNA-binding specificity of RBM39 remains to be addressed experimentally.

To decipher the role of RBM39 in RNA metabolism, we combined functional and structural approaches. Our data revealed that RBM39 actively participates in splice site selection and autoregulates through a negative feedback loop by enhancing the inclusion of a poison exon in its own pre-mRNA. This splicing regulation involves all three RRM domains as well as the RS domain along with two *cis*-acting sequence elements in the RBM39 pre-mRNA. Furthermore, we decipher the molecular basis of RBM39-RNA recognition by its RRM1 and RRM2. Our data bring molecular insights into RBM39-dependent 3′-splice site selection by this cancer-associated splicing factor.

## Results

### RBM39 autoregulates its expression by alternative splicing

To investigate the role of RBM39 in RNA metabolism, we performed RNA-Seq in HeLa cells treated with either control (Ctrl KD) or RBM39 siRNAs (RBM39 KD) and rescued the RBM39 KD by co-transfecting RNAi-resistant FLAG-RBM39. The principal component analysis confirmed a strong clustering of the three biological replicates (Supplementary Fig. 1). We then performed differential expression analysis using DESeq2[33] and computed meta p-values comparing both Ctrl versus KD and rescue versus KD[34]. Using this approach, we found 7,233 mRNAs whose steady-state levels were affected by RBM39 (Fig. 1a). Next, we used DEXseq[35] to identify 11,134 alternative exon skipping and 7275 intron retention events that were altered in an RBM39-dependent manner (Fig. 1b, c). Among the 1000 most significant events, 73.3% of the transcriptional changes, 82.7% of the exon skipping events and 93.8% of the intron retention events were rescued, indicating that the alterations are not caused by off-target effects of the RBM39 siRNAs. Loss of RBM39 induced both up- and down-regulation of mRNAs, as well as increased inclusion and skipping of alternative exons. In contrast, more than 80% of the intron retention events showed decreased splicing efficiencies upon RBM39 KD, indicating that RBM39 predominantly enhances the efficiency of constitutive splicing. The transcriptomic data also revealed that RBM39 promotes the inclusion of a cassette exon (exon 2b) into its own mRNA, which produces an unproductive isoform with a premature termination codon (PTC) that is predicted to be degraded by the nonsense-mediated decay (NMD) machinery[36] (Fig. 1d, e). Such PTC-containing exons are sometimes referred to as poison exons. Importantly, this alternative splicing pattern was not caused by the lower levels of RBM39 mRNA in the KD condition, as the expression of FLAG-RBM39 restored the inclusion of exon 2b (Fig. 1f). To assess if the PTC-containing isoform is indeed targeted by NMD, we knocked down the central NMD factor UPF1 in HeLa cells using siRNAs. This treatment severely reduced UPF1 mRNA and protein levels and induced a 20-fold upregulation of the endogenous NMD substrate RP9P[34]. Total RBM39 mRNA levels were increased about 3-fold upon UPF1 KD, suggesting that in HeLa cells about 66% of the transcripts are degraded by the

NMD pathway under physiological conditions (Supplementary Fig. 1). To validate our findings in an independent and clinically relevant dataset, we then explored the alternative splicing of RBM39 pre-mRNA in RNA-Seq data from The Cancer Genome Atlas (TCGA) using TCGASpliceSeq[37]. Previous reports have shown that in this dataset, total RBM39 levels are highest in AML compared to other cancer subtypes[8]. Consistent with the role of RBM39 in promoting poison exon inclusion, we found that the median inclusion efficiency of exon 2b was also highest in AML (Fig. 1g). Thus, we conclude that the autoregulation mechanism operates in human tumour samples. Altogether, our data show that RBM39 tightly autoregulates its level of expression at the pre-mRNA splicing stage using a negative feedback loop mechanism.

### All four RBM39 domains contribute to RBM39 function

To support the transcriptomic data, we validated three representative RBM39-dependent intron retention events observed in three pre-mRNA targets (*MBD1*, *TPP1* and *PAPOLA*, see Supplementary Fig. 2a) and investigated the functional contribution of each RRM. Hereto, we rescued RBM39 KD with either FLAG-tagged wild-type RBM39 or mutants lacking one RRM each (Fig. 2a, b). All constructs were moderately overexpressed and none of the deletions affected the RBM39 subcellular localisation (Fig. 2c and Supplementary Fig. 2b). In agreement with our transcriptomic data, RBM39 KD perturbs the splicing of the three introns while the rescue using the wild-type construct either partially or fully restored the splicing defects, indicating that the candidates differ in their sensitivities towards altered RBM39 levels. In particular, TPP1 pre-mRNA splicing may require higher amounts of RBM39 since multiple consecutive introns are retained upon RBM39 depletion (Supplementary Fig. 2a). However, for all three intron retention events, the mutant lacking RRM1 (ΔRRM1) did not retain any function, indicating an important role of this domain. Furthermore, the mutants lacking RRM2 (ΔRRM2) or RRM3 (ΔRRM3) only partially rescued the splicing defects compared to the wild-type protein, suggesting that they are also involved in the splicing mechanism (Fig. 2d). Interestingly, the functionality profile of the constructs observed for the intron retention events also holds true for alternative splicing of cassette exons, as assessed by the inclusion of the poison exon in the RBM39 mRNA (Fig. 2e–g). In summary, all three RRM domains are involved in RBM39-dependent splicing, with a major role played by RRM1.

To investigate the role of the RS domain in RBM39-dependent splicing, we carried out rescue experiments using N-terminally truncated FLAG-tagged RBM39 fused to a heterologous SV40 nuclear localisation signal (Fig. 3a, b). Compared to wild-type RBM39, which robustly associates with nuclear speckles, the ΔRS mutant was diffusely localised in the nucleus (Fig. 3c, d), indicating a role of the RS domain in speckle recruitment. Moreover, the ΔRS mutant was unable to rescue the three intron retention events as well as the inclusion of exon 2b into RBM39 mRNA (Fig. 3e–g). Therefore, we conclude that the RS domain is essential for RBM39's role in splicing.

### RBM39 interacts with early spliceosome components

RBM39 and its yeast homolog were previously identified in early spliceosome complexes and protein–protein interactions with the U2 snRNP component SF3B1 as well as the U1 snRNP associated factor U1-70K have been described[23–25,29,38,39]. Thus, we first aimed to independently verify these interactions using co-immunoprecipitation in HeLa nuclear extracts. First, all Sm-class U snRNPs were immunoprecipitated with an anti-Y12 antibody[40]. The isolated complexes contained RBM39 and the U1 snRNP-specific protein U1-C (Fig. 4a), confirming that RBM39 interacts with the spliceosome. We then precipitated U1 snRNP with an anti-U1A antibody, which again pulled down RBM39 and U1-C (Fig. 4b), confirming that RBM39 associates with the U1 snRNP. However, while the U2 snRNP protein SF3A3 co-precipitated with

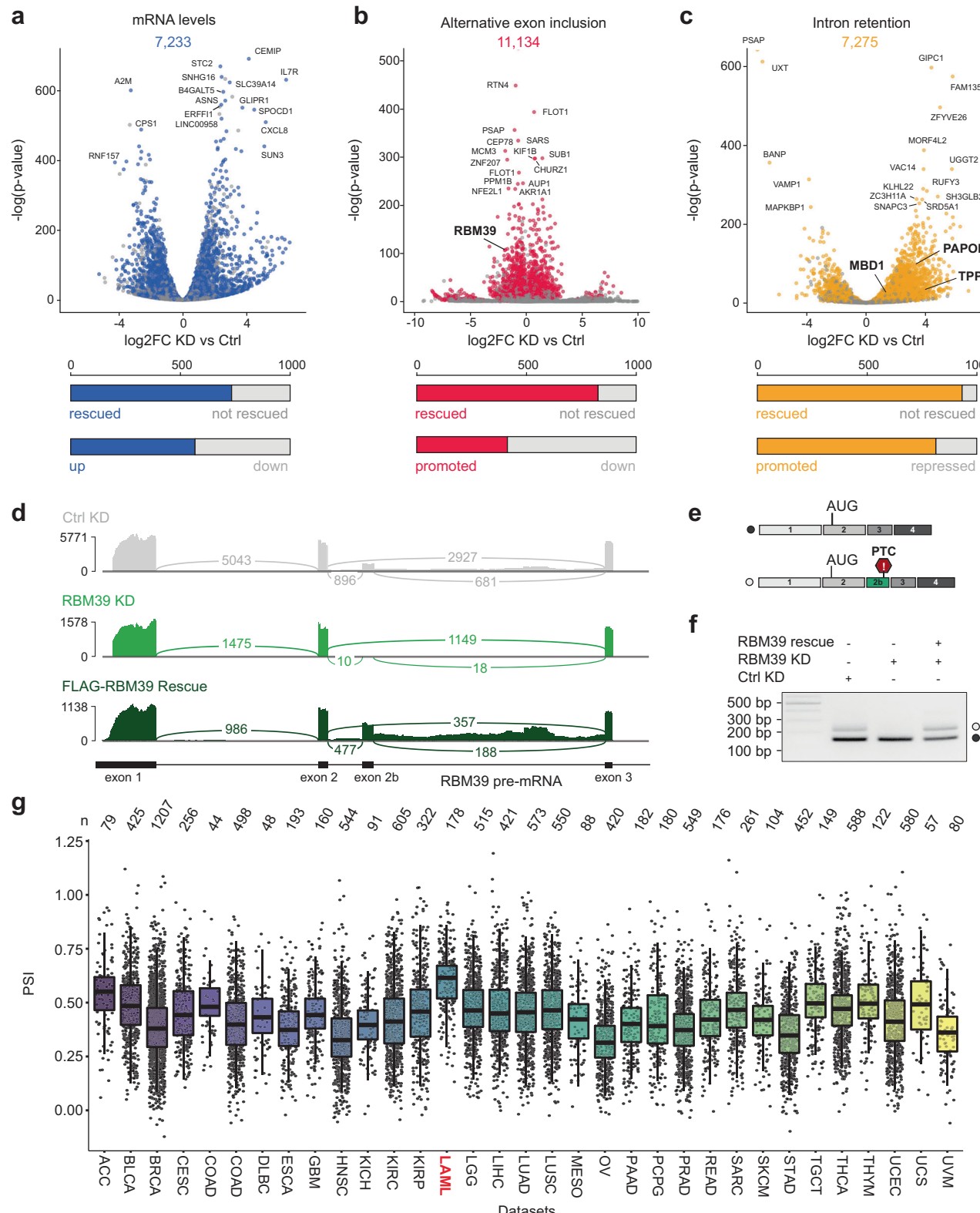

endogenous RBM39 in an RNA-independent manner, co-precipitation of U1-A was sensitive to RNA digestion (Fig. 4c). Therefore, our results do not support a direct contact between RBM39 and U1-70K and rather suggest that the interaction with U1 snRNP could be indirect or mediated via the U1 snRNA.

To identify the regions of RBM39 involved in contacting the splicing machinery, we conducted IP experiments using the previously mentioned FLAG-tagged RBM39 deletion mutants lacking either the

RS domain, the RRM1, RRM2, or RRM3 domains upon transient transfection into HeLa cells. Total cell extracts were prepared in a low-salt buffer containing 0.5% Triton-X-100, which previously allowed us to study the interaction between the splicing factor FUS and U1 snRNP[41]. Under these conditions, we found that RBM39 co-precipitated SF3A3 but was unable to pull down U1-C (Fig. 4d), arguing against a robust interaction between RBM39 and the U1 snRNP. While deletion of RRM1 or RRM2 did not significantly affect

**Fig. 1 | RBM39 controls the inclusion of a poison exon into its own mRNA.**
**a**–**c** Volcano plots showing up- and downregulation of transcripts (**a**), alternative exons (**b**), and retained introns (**c**), upon RBM39 depletion. *P* values were computed using two-sided Wald test (**a**, **c**) or $\chi^2$ likelihood-ratio test (**b**), and adjusted for multiple comparison using the procedure of Benjamini and Hochberg, *n* = 3. Statistically significant alterations ($p_{adj}$ <0.05) are highlighted in colour. The bar plots indicate the fraction of rescued events ($p_{adj}$ <0.05) and their directionality among the 1000 most significant changes. **d** Sashimi plot showing the reads observed for the beginning of the *RBM39* gene upon control knock down (Ctrl KD), RBM39 knock down (RBM39 KD) and FLAG-RBM39 rescue. **e** Schematic representation of both RBM39 mRNA isoforms. In the longer isoform, the exon 2b is included and introduces a premature termination codon (PTC). **f** RT-PCR validation of the RBM39-dependency of exon 2b inclusion. Quantification and statistics are shown in Fig. 2g. **g** Boxplot showing the percent of exon 2b inclusion in transcriptomic data grouped according to cancer type. This plot was generated using RNA-Seq data from The Cancer Genome Atlas (TCGA). Within each box, horizontal black lines denote median values. Lower and upper hinges correspond to the 25th and the 75th percentile, respectively, whereas vertical lines extend to the most extreme values within 1.5 × interquartile range. For cancer type, n is indicated in the plot.

the interaction between RBM39 and SF3A3, this interaction was perturbed when the RRM3 was mutated (Fig. 4d), consistent with previous observations[23–25]. Surprisingly, we found that the interaction with SF3A3 was also lost when the RS domain of RBM39 was deleted, indicating that both RRM3 and the RS domain of RBM39 are important for its association with U2 snRNP (Fig. 4d, e). In summary, our findings suggest that RBM39 interacts directly with U2 snRNP through RRM3 and the RS domain, while its association with U1 snRNP is weak, RNA-dependent and may occur indirectly in assembled pre-spliceosomal complexes.

## RBM39 RRM2 binds single-stranded RNA motifs using an extended RNA-binding interface

To identify the main mRNA-binding domain of RBM39, we performed RNA immunoprecipitation (RIP) using FLAG-tagged RBM39 or mutants where either RNA-binding by RRM1 or RRM2 or the UHM-ULM interactions of RRM3 were disrupted by point mutations (detailed mutations are illustrated, see below) and measured the levels of co-precipitating mRNA by RT-qPCR. All bait proteins were purified with comparable efficiencies and have rich protein interactomes. Compared to a no-transfection control (NTC) condition, the RBM39-regulated mRNAs MBD1, PAPOLA and TPP1 as well as the housekeeping mRNAs GAPDH, HSP90 and TUBA1A were significantly enriched by FLAG-RBM39. However, we did not observe apparent differences between the mutant constructs, suggesting that RBM39 interacts with mRNPs in a redundant manner involving both protein–protein as well as protein–RNA contacts (Supplementary Fig. 3). We therefore repeated the experiment using FLAG-RBM39 constructs encompassing only RRM1 and RRM2 followed by a heterologous SV40 nuclear localisation signal. Under these simplified conditions, the WT and RRM1 mutant proteins retained their rich protein interactomes and robustly enriched the housekeeping mRNAs (Fig. 5a). Surprisingly, we did not detect an enrichment of the RBM39-regulated mRNAs with this minimal RBM39 constructs. However, the RRM2 and RRM12 mutants lost their ability to bind any mRNA and consequently display strongly reduced protein interactomes, indicating that RRM2 is the main mRNA-binding interface of RBM39. Using NMR spectroscopy, we tested the binding of RRM12 (a construct encompassing RBM39 RRM1 and RRM2) in vitro on different single-stranded RNA (ssRNA) motifs that were previously isolated by CLIP[32]. In line with the RIP experiments, the CLIP-derived sequences mainly induced amide CSPs on the resonances of RRM2 and the binding became optimal when RRM12 was titrated with 5′-AGCUUUG-3′ (Supplementary Fig. 4). A similar observation was made using isolated RRM2 which binds 5′-AGCUUUG-3′ with a $K_d$ of 9 ± 2 µM (Supplementary Fig. 4 and Table 1) and induces large amide CSPs when the interaction was monitored by NMR spectroscopy (Fig. 5b). Saturation of the CSPs required a slight excess of RNA (1.5-fold excess) and the plots of the amide and carbonyl CSPs as a function of the protein sequence highlighted three main areas of contacts: the loop β1-α1, both β1 (RNP2) and β3 (RNP1) and the C-terminal tail (Fig. 5b). The solution structure of RRM2 in complex with 5′-AGCUUUG-3′ was solved using 2361 NOE-derived distances including 84 intermolecular distances (Fig. 5c, Supplementary Fig. 5, Supplementary Video 1 and Supplementary Table 2). The 20 NMR structures overlaid with a backbone RMSD of 0.45 Å and show that RRM2 specifically recognises two distinct RNA patches (Fig. 5c, d). On the β-sheet surface, the 3′-dinucleotide $U_6$-$G_7$ stacks on the aromatic residues (F295 and Y253) and established direct hydrogen bonds with the backbone of the C-terminal extremity of the protein (Fig. 5e). Both preceding nucleotides $U_4$ and $U_5$ are directly recognised by K322 and R329, respectively, explaining the specificity for the motif UUUG. At the 5′-end, the $A_1$-$G_2$-$C_3$ trinucleotide interacts with the β1-α1 and β2-β3 loops. A key feature of the interaction is the insertion of F259 between $A_1$ and $G_2$ and the stabilisation of both purines by positively charged amino acids on each side, namely R289 and H258 (Fig. 5f). Mutation of F259 into alanine reduced the binding affinity 4-fold, in line with the structure (Table 1). Furthermore, the structure revealed a direct hydrogen bond between the O6 atom of $G_2$ and the backbone amide group of N260 as well as between the N3 amino of $C_3$ and the amide of L256. In agreement, the amide groups of L256 and of N260 are strongly downshifted by the addition of ssRNA. The 5′-terminal A stacks between F259 and R289 and its specific recognition is achieved by the formation of a direct hydrogen bond between the N6 amino and the side chain hydroxyl oxygen of S290. In agreement with the structure, mutations of F259, R289 and H258 to alanine strongly reduced the RNA-binding affinity of RRM2, supporting the important role of the extended interface to achieve high affinity for ssRNA (Table 1). Altogether, the solution structure of RRM2 bound to ssRNA revealed an extended RNA-binding interface that combines the β-sheet surface as well as the α1-β1 and β2-β3 loops and enables RBM39 to select target pre-mRNA.

## RBM39 RRM1 recognises the shape of RNA stem loops

In sharp contrast to RRM2, RRM1 did not bind any single-stranded RNA (ssRNA) motifs that were previously isolated by CLIP[32]. In the course of studying interactions between RBM39 and the splicing machinery, we tested a potential RNA-dependent direct interaction between in vitro reconstituted U1 snRNPs[42,43] and an ILV $^{13}$C-methyl-labelled RBM39 RRM12. During this titration, we observed changes of the ILV methyl chemical shifts of RRM12 and more particularly those located in RRM1 (Supplementary Fig. 6). The structure of U1 snRNP[44] revealed two protein-free and solvent exposed stem loops (SL3 and SL4) which represent major hubs for the communication with splicing factors[41,43,45–47]. Interestingly, similar ILV methyl chemical shifts perturbations (CSP) of RRM12 were observed when the protein was titrated with isolated U1 snRNA SL3 or SL4 (Supplementary Fig. 6), suggesting that RRM1 could contact either SL3 or SL4 when bound to U1 snRNP in vitro. These observations were further validated when $^{15}$N-labelled RRM12 was titrated with either in vitro transcribed SL3 or SL4 (Fig. 6a). To test if RRM1 has an intrinsic preference for U1 snRNA stem loops, we repeated the experiment with an unrelated stem loop that was previously identified as an aptamer of RBMY[48]. As similar chemical shift perturbations were observed for all three stem loops, we conclude that RBM39 RRM1 interacts with RNA stem loops with no strong sequence preference (Fig. 6a). Using isothermal titration

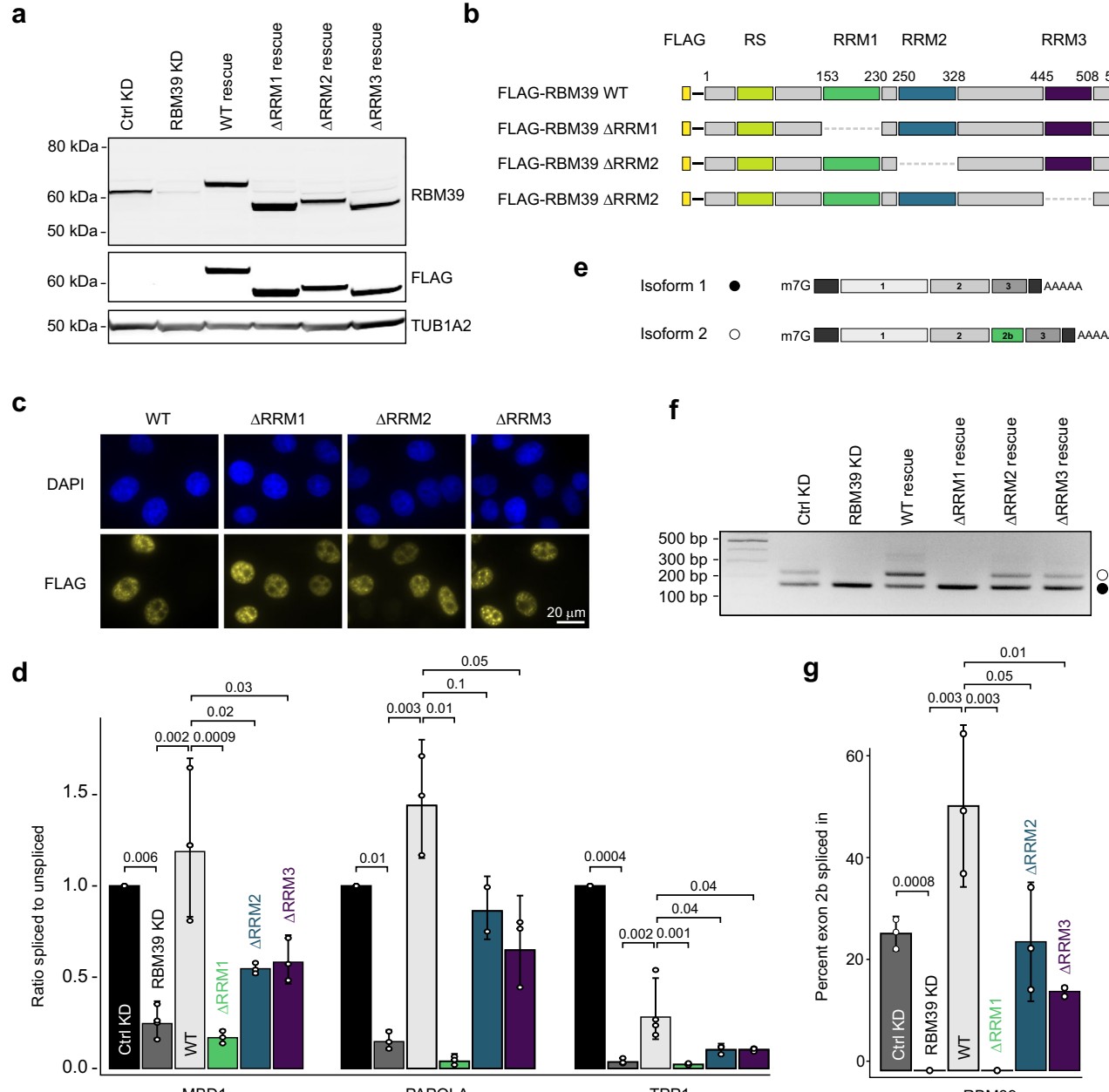

**Fig. 2 | Contributions of the three RBM39 RNA recognition motifs to splicing regulation. a** Western blot analysis of RBM39 levels upon control knockdown (Ctrl KD), RBM39 knockdown (RBM39 KD), FLAG-RBM39 rescue (WT rescue) or rescue performed with RBM39 lacking each one RRM (ΔRRM1, ΔRRM2 and ΔRRM3 rescues). HeLa cell extracts were subjected to SDS–PAGE and Western blotting with anti-RBM39 and anti-FLAG antibodies. Tubulin 1A2 served as a loading control. **b** Schematic representation of the FLAG-tagged RBM39 deletion constructs. **c** Immunofluorescence analysis of FLAG-RBM39 constructs upon transient expression in HeLa cells. Exogenous RBM39 was visualised using anti-FLAG antibodies and nuclei were counterstained with DAPI, n = 3. **d** RT-qPCR measurements displayed as the ratio of spliced to unspliced isoform for three intron retention

events identified by RNA-Seq (MBD1, PAPOLA and TPP1). Average values and standard deviations of three (ΔRRM1, ΔRRM2, ΔRRM3) or four (Ctrl, RBM39 KD, WT) biological replicates (n = 3 or 4) are shown. P values were computed from log-transformed ratios using two-sided unequal variances Welch's t test[65]. **e** Schematic representation of both RBM39 mRNA isoforms. In the longer isoform, the exon 2b is included and introduces a premature termination codon (PTC). **f** Agarose gel showing the effect of RRM deletion mutants on the splicing of the poison exon of RBM39 mRNA as assessed by RT-PCR. **g** Barplot showing the percent of exon 2b inclusion in the different conditions. Average values and standard deviations of three biological replicates (n = 3) are shown. P values were computed using two-sided t test.

calorimetry (ITC), we determined that RRM1 binds to SL3 with a $K_d$ of $15 \pm 3 \mu M$ and to SL4 with a $K_d$ of $11 \pm 4 \mu M$ (Table 1).

In order to decipher the atomic details of the RBM39 RNA stem-loop recognition, we studied the interaction between RRM1 and the U1 snRNA stem loop 3 using NMR spectroscopy. Upon addition of SL3 into a sample of $^{15}N$-labelled RRM1, we observed amide CSPs in an unusually large part of the RRM which covers β1 (RNP2), the β1−α1

loop (near A161), the β2-β3 loop (near R192), β3 (RNP2), and the β4-β5 loop (near G220) (Fig. 6b). On the RNA side, the NMR signals from the loop were the most affected (Supplementary Fig. 7). The solution structure of RRM1 bound to SL3 was determined using 2576 NOE-derived distances including 66 intermolecular restraints (Supplementary Table 1, Supplementary Video 2 and Supplementary Fig. 7). The ensemble of structures overlaid with a backbone root mean

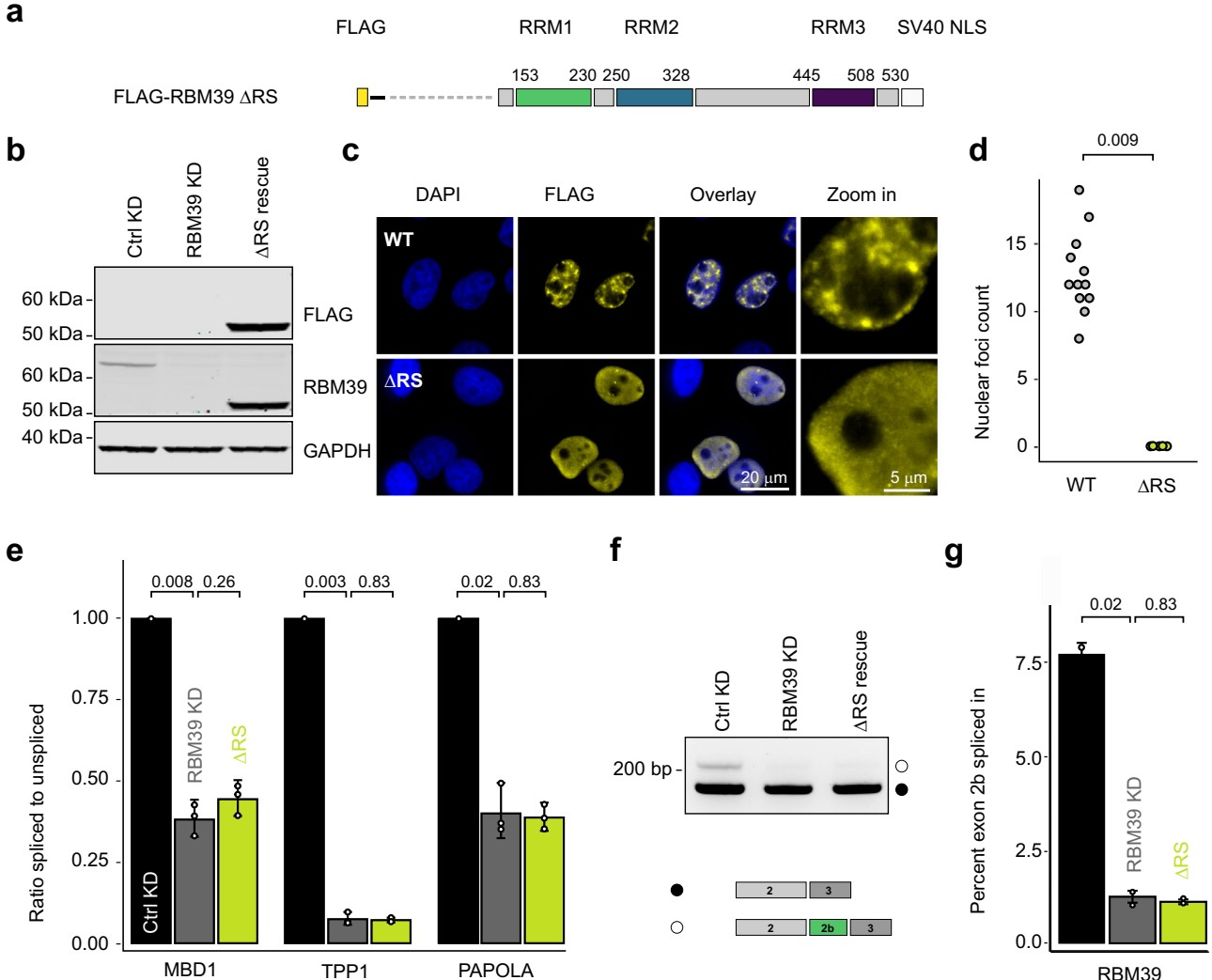

**Fig. 3 | The RS domain of RBM39 is essential for RBM39 function in splicing.**
**a** Schematic representation of the FLAG-tagged RBM39 construct lacking the RS domain. **b** Western blot analysis of RBM39 levels upon control knockdown (Ctrl KD), RBM39 knockdown (RBM39 KD), and rescue with FLAG-RBM39 lacking the RS domain (ΔRS). HeLa cell extracts were subjected to SDS–PAGE and Western blotting with anti-RBM39 and anti-FLAG antibodies. GAPDH served as a loading control. $n = 3$. **c** Immunofluorescence analysis of FLAG-RBM39 constructs upon transient expression in HeLa cells. Exogenous RBM39 was visualised using anti-FLAG antibodies and nuclei were counterstained with DAPI. **d** Plot showing the number of nuclear foci counts of FLAG-RBM39 and FLAG-RBM39 ΔRS. $P$ values were computed using two-sided $t$ test, $n = 3$. **e** RT-qPCR measurements displayed as the ratio of spliced to unspliced isoform for three intron retention events identified by RNA-Seq (MBD1, PAPOLA and TPP1). Average values and standard deviations of three biological replicates ($n = 3$) are shown. $P$ values were computed from log-transformed ratios using two-sided unequal variances Welch's $t$ test[65]. **f** Agarose gel showing the effect of RS domain deletion on the splicing of the poison exon of RBM39 mRNA as assessed by RT-PCR. Below the gel, schematic representation of both RBM39 mRNA isoforms. **g** Barplot showing the percentage of exon 2b inclusion in the different conditions. Average values and standard deviations of three biological replicates ($n = 3$) are shown. $P$ values were computed using two-sided $t$ test.

square deviation (RMSD) of 0,68 ± 0,17 Å and revealed that RRM1 interacts with three bases in the RNA loop using its β-sheet surface ($A_{104}$, $U_{105}$ and $G_{106}$). The interactions with $U_{105}$ and $G_{106}$ are only mediated by stacking interactions against the aromatic rings of F156 and Y198 while $A_{104}$ inserts into a cavity along β5 and interacts directly with the side chain of Q159 (Fig. 6c–e). The main interaction surface is mediated by the β2-β3 loop that interacts with the major groove adjacent to the loop. This β2-β3 loop contains four basic side chains (R188, R191, R192 and K194) that establish direct contacts with the phosphate backbone of the loop and with the two GC base pairs at the apical part of the stem (Fig. 6f). There are additional interactions with the RNA stem, as the β1−α2 and the β4−β5 loops establish polar interactions with the RNA backbone. The structure of the protein−RNA complex is in total agreement with the amide CSPs observed upon the addition of SL3. Its analysis revealed that RRM1

has a poor sequence specificity and rather recognises the shape of this RNA stem loop. The structure of RRM1 bound to SL3 revealed a large positive surface, which perfectly accommodates the RNA loop and its adjacent major groove (Supplementary Fig. 8). To conclude, the structure of RBM39 RRM1 bound to SL3 explains how RRM1 recognises the RNA stem-loop shape.

**Functional validation of the RBM39−RNA interfaces**
In order to test the functional relevance of our protein−RNA complex structures, we mutated key residues at the protein/RNA interfaces and tested the ability of mutants to rescue RBM39 KD on the three RBM39-dependent intron retention events. We designed a mutant altering RNA-binding by RRM1 on the β-sheet surface (mRRM1.1; F156A/Y198A/R192A), and one mutant altering the contact from the basic residues of the loop β2-β3 (mRRM1.2; R188A/R191A/R192A/K194A). We also

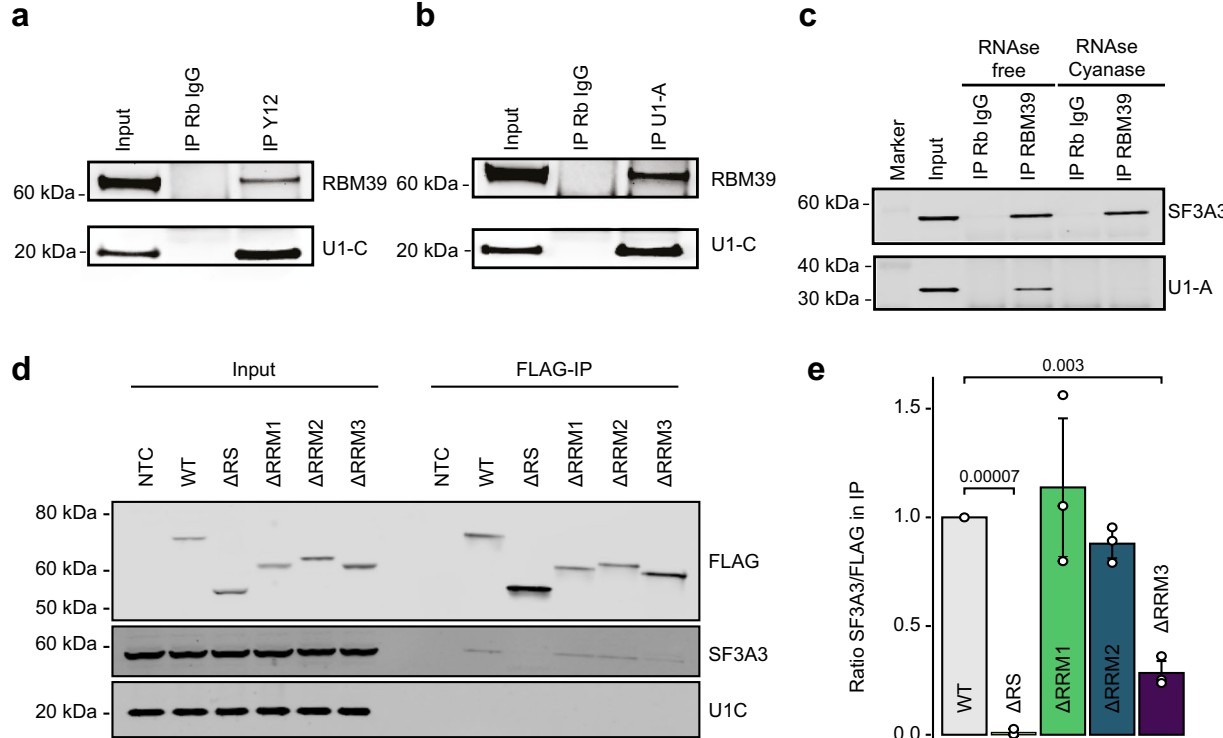

**Fig. 4 | RBM39 interacts with spliceosomal components. a** Immunoprecipitation from HeLa nuclear extracts performed using an anti-Y12 antibody to pull down all Sm-class snRNPs. The western blots were probed with anti-RBM39 and anti-U1C antibodies ($n = 3$). **b** Immunoprecipitation from HeLa nuclear extracts performed using an anti-U1A antibody to pull down U1 snRNP. The western blots were probed with anti-RBM39 and anti-U1C antibodies ($n = 3$). **c** Immunoprecipitations from HeLa nuclear extracts performed using an anti-RBM39 antibody in absence or presence of RNAse and Cyanase. Western blots were probed with antibodies against SF3A3 and U1-A ($n = 3$). **d** FLAG immunoprecipitation of different FLAG-RBM39 constructs upon transient transfection in HeLa cells. Western blots were probed using anti-SF3A3 and anti-U1-C antibodies ($n = 3$). **e** Barplot showing the ratio between the western blot signal intensities of SF3A3 versus FLAG in the IP fractions. Average values and standard deviations of three biological replicates ($n = 3$) are shown. *P* values were computed using two-sided *t* test.

prepared an RBM39 mutant altering the RNA-binding interfaces of RRM2 (mRRM2; Y253A/H258A/F259A/F295A), another altering the interaction of RRM3 with U2 snRNP (mRRM3; R494A/F496A) and finally a mutant combining mutations of both RRM2 and RRM3 (mRRM2&3) (Fig. 7a). All the proteins expressed at similar levels (Fig. 7b), localised in the nucleus (Fig. 7c). The RRM mutants were also produced recombinantly, and their correct folding was confirmed using NMR spectroscopy (Supplementary Fig. 9). In agreement with the structural data, the RRM mutants have a strongly reduced affinity for their respective targets: mRRM1.1 and mRRM1.2 have a reduced affinity for SL3, mRRM2 has a reduced affinity for AGCUUUG and finally mRRM3 does not bind to SF3b155 ULM (Table 1 and Supplementary Fig. 9). Compared to wild-type RBM39, none of the mutants fully rescued the RBM39 KD in all three intron retention events. The mutants showing complete loss of function were mRRM1.1, mRRM1.2 and mRRM2&3 (Fig. 7d). Analogous results were obtained for the RBM39 autoregulation splicing event that involves the inclusion of the poison exon 2b (Fig. 7e). Altogether, these functional results strongly support our structural findings involving protein−RNA interactions for RRM1 and RRM2 and protein−protein interactions for RRM3.

## RBM39 autoregulates its splicing through non-canonical 3′-splice site selection

To identify the *cis*-acting RNA elements responsible for RBM39-dependent splicing, we first inserted the poison exon and approximately 100 nucleotides of its flanking intronic regions into the human β-globin (HBB) gene between exons 2 and 3. In this heterologous context, the RBM39-dependency of poison exon inclusion

was preserved, indicating that the *cis* regulatory elements are located in proximity of the poison exon and its flanking regions (Supplementary Fig. 10). We therefore analysed the sequence of the poison exon and identified two potential binding sites for RRM2 on the basis of the structure: CUCUUUG (BS1) and ACCUUUG (BS2) (Fig. 8a). While BS1 is located immediately upstream of the AG dinucleotide of the 3′-splice site, BS2 is located within the exon and could potentially form an inhibitory stem loop structure (SLi) by pairing with the 5′-splice site of the poison exon (Fig. 8a). To investigate the role of these putative binding sites, we used our RBM39 minigene encompassing exon 1 to 4 and either deleted BS2 (ΔBS2), substituted the intronic sequence at the 3′-splice site with the β-globin 3′-ss (HBB 3′ ss) or specifically mutated BS1 (mutBS1). The effects of the mutations were evaluated on the splicing of the poison exon upon Ctrl or RBM39 KD (Fig. 8b–d). Compared to the endogenous RBM39 mRNA, poison exon inclusion was more efficient in the minigene (70%), likely because the minigene mRNA is less susceptible to nonsense-mediated decay[36]. However, upon RBM39 KD, the percentage of poison exon inclusion in the minigene dropped to zero. In the ΔBS2 minigene, exon inclusion in the Ctrl KD condition was significantly increased compared to the wild-type minigene which could be explained in a scenario where deletion of BS2 disrupts the predicted inhibitory stem loop (SLi) and thus activates poison exon inclusion in an RBM39-independent manner. Nevertheless, RBM39 KD completely abolished poison exon inclusion suggesting that BS2 is not the primary binding site for RBM39. In agreement, the stabilisation of SLi (OPT-SLi) reduced poison exon inclusion (Fig. 8d). Then, we substituted the region upstream of the poison exon with the 3′-ss sequence of a constitutively spliced HBB exon (Fig. 8b and

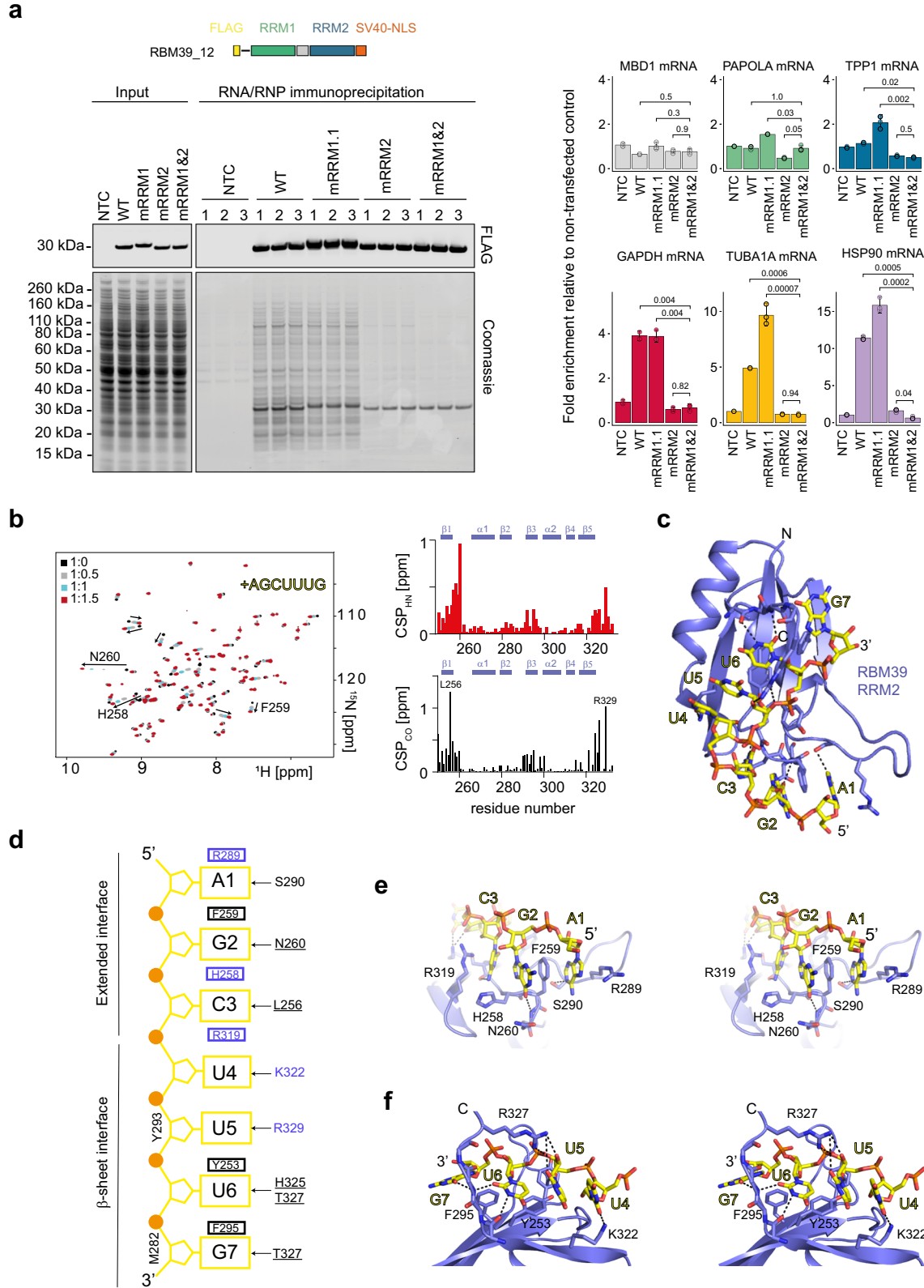

Supplementary Fig. 10). In this genetic context, baseline exon inclusion increased to 100% and the RBM39-dependency of the poison exon was lost. Finally, we specifically substituted BS1 by a CCUCCCA motif as in the HBB 3′ss construct (Fig. 8b and Supplementary Fig. 10). This subtle change was sufficient to render the poison exon inclusion RBM39-independent. Importantly, we could replicate these findings by inserting the identical mutations at BS1

and BS2 into the chimeric HBB-RBM39 minigene, confirming that the observed effects are independent of the minigene context (Supplementary Fig. 10). In line with our functional data, we confirmed using NMR spectroscopy that RRM2 binds strongly to BS1 but not to mutBS1 (Fig. 8c). Altogether, our experiments uncovered two *cis*-acting sequence elements which govern poison exon inclusion and among those, one is essential for being RBM39-dependent.

**Fig. 5 | Structural basis for RBM39 single strand RNA-binding activity. a** RNA/RNP immunoprecipitation using FLAG-RBM39 RRM12. On the left side, a scheme of FLAG-RBM39 RRM12 is shown. Below, a western blot probed using anti-FLAG antibodies indicates similar expression of the constructs in the input and confirms efficient immunoprecipitation. The gel below was stained using Coomassie blue. On the right, the amount of co-immunoprecipitated mRNAs was quantified using RT-qPCR. We detected the three mRNA targets (MBD1, PAOLA and TPP1) of RBM39 and three housekeeping mRNAs (GAPDH, HSP90 and TUB1A). Average values and standard deviations of three biological replicates ($n = 3$) are shown. $P$ values were computed from log-transformed ratios using two-sided unequal variances Welch's $t$ test[65]. **b** Overlay of the 2D $^{15}$N-$^1$H HSQC spectra of $^{15}$N-labelled RRM2 recorded upon successive stepwise additions of the 5'-AGCUUUG-3' ssRNA motif. Spectra are coloured according to the molar ratio protein:RNA (black 1:0; grey 1:0.5; cyan 1:1; red 1:1.5). The titration was performed using a 200 µM protein solution and a 2 mM RNA stock solution. Normalised amide and carbonyl CSPs are plotted as a function of the protein sequence. **c** Representation of the lowest energy model of the solution structure of the RRM2–AGCUUUG complex. **d** Schematic representation of the protein–RNA contacts observed in the solution structure of the RRM2–AGCUUUG complex. **e** Stereo view of the β-sheet RNA-binding interface. **f** Stereo view of the extended RNA-binding interface involving the loop β1-α1.

## Discussion

### RNA recognition by RBM39

The solution structures of both N-terminal RRMs of RBM39 bound to their respective RNA targets revealed that RRM1 has a strong preference for RNA stem-loop structures while RRM2 specifically interacts with 5'-N(G/U)NUUUG-3' motifs. When compared to the RRMs of FUS and RBMY, which are also specific for RNA stem loops[41,48,49], the structure of RRM1 bound to SL3 revealed a common strategy used by the three RRMs for the recognition of the stem loop shape. All three RRMs use their β-sheet surface to interact with bases of the loop and insert a loop (β2-β3 in the case of RBM39 and RBMY and β1-α1 in the case of FUS) in the adjacent major groove (Supplementary Fig. 8). In contrast, RRM2 anchors the splicing regulator to the pre-mRNPs, as shown by the RIP experiments. The structure of RRM2 bound to its ssRNA target uncovered an extended RNA-binding interface with contacts to seven nucleotides that combines the β-sheet surface and the loops β1-α1 and β2-β3 of the domain. A similarly extended RNA-binding interface was found earlier in Rb-Fox1 RRM; however, the RNA-binding affinity of the Rb-Fox1 RRM for UGCAUGU is 18,000 times stronger[50]. This large difference could be explained by the formation of an intramolecular RNA base pair between $G_2$ and $A_4$ and a larger network of intermolecular hydrogen bonds in the case of Rb-Fox1 (Supplementary Fig. 5). Although the binding affinities of both RRMs for their cognate RNA elements are in the micromolar range, the combination of both RNA elements on the same molecule conferred much higher affinity for RBM39 (20 to 30-fold increase, Supplementary Fig. 4). As observed in the case of FUS, RBM39 could bind RNA targets with a bipartite RNA recognition mode[49]. This property could

be used to select bipartite high-affinity motifs on pre-mRNA targets or to link two RNA molecules.

### Molecular mechanisms of RBM39 autoregulation

Among the RBM39-dependent cassette exons, a PTC-containing exon-inducing mRNA decay was identified in the *RBM39* gene. Such a negative feedback loop mechanism is commonly observed in autoregulation of key splicing or transcription regulators[51–54], allowing a precise control of critical gene expression regulator homoeostasis. Our experimental data reveal that the inclusion of the poison exon is controlled by two distinct mechanisms: one RBM39-dependent and one RBM39-independent.

First, a 5'-CUCUUUG-3' motif at the 3'-splice site is essential for alternative splicing of the poison exon as mutations in this motif abolish RBM39-dependency and strongly increase baseline exon inclusion. Given that this 5'-CUCUUUG-3' motif perfectly matches the 5'-N(G/U)NUUUG-3' consensus motif of RRM2 inferred from our structural studies, we propose that RBM39 selects the 3'-splice site using RRM2 and stabilises the U2 snRNP on the branchpoint by interacting with ULM motifs in SF3b155 via its RRM3 (UHM) domain. In principle, RBM39 could also recruit the ULM-containing proteins U2AF2 or SF1 to the 3'-splice site; however, the affinity of RBM39 for SF3b155 is at least 20-fold higher compared to these other proteins[25]. Our data also support an important role of the RS domain in stabilising the association between RBM39 and U2 snRNP. A possible explanation for the increase in baseline exon inclusion is that the mutated 3'-splice site that originates from a constitutive exon in the β-globin gene is recognised more efficiently by U2AF2 which may functionally substitute for RBM39 by recruiting U2 snRNP (as illustrated in Fig. 9). In line with this proposition, RBM39 was previously shown to associate with U2AF1[29]. Nevertheless, we cannot formally exclude that the 3'-splice site mutations prevent the association with an unknown negative splicing regulator.

Second, an inhibitory stem loop (SLi) acts as a repressive element by sequestering the 5'-splice site of the poison exon. Similar structures have been previously described to regulate alternative splicing of cassette exons such as *SMN2* exon 7[55] or *Map/Tau* exon 10[56]. However, this *cis*-acting RNA element modulates poison exon splicing in an RBM39-independent manner. Upon mutation of the inhibitory stem loop that releases the 5'-splice site, we observed an increase baseline exon inclusion that was reset by RBM39 depletion (Fig. 8). Our functional assays support an important role played by RRM1 for RBM39's function in splicing (Fig. 2). In this context, we propose two scenarios that could be compatible with our experimental data. In the context of a pre-spliceosomal complex, the stabilisation of U1 snRNP on this weak 5'-splice site could be mediated via a weak interaction between RRM1 and a U1 snRNA stem loop. This weak interaction could be strengthened by additional RS/RS contacts[29]. In agreement, direct interactions between splicing factors and the U1 snRNA stem loops SL3 and SL4 are frequently observed[47]. Another possible scenario is that RBM39 RRM1 interacts with SLi and reinforces the anchoring of RBM39 to the pre-mRNA. In line with this later scenario, we could observe using NMR

## Table 1 | Thermodynamic analysis of the protein–RNA and protein–protein interactions

| Protein | RNA | $K_d$ (µM) | $n$ | ΔH (kcal/mol) | −TΔS (kcal/mol) |
|---|---|---|---|---|---|
| RBM39 RRM1 | SL3 | 15 ± 3 | 0.95 | −136 | 108 |
| RBM39 RRM1 | SL4 | 11 ± 4 | 0.78 | −68 | 40 |
| RBM39 mRRM1.1 | SL3 | >100 | n.d. | n.d. | n.d. |
| RBM39 mRRM1.2 | SL3 | >100 | n.d. | n.d. | n.d. |
| RBM39 RRM2 | AGCUUUG | 9 ± 2 | 1.18 | −8 | −0.8 |
| RBM39 RRM2 F259A | AGCUUUG | 41 ± 10 | 1.27 | −0.5 | 0.9 |
| RBM39 RRM2 H258A/F259A/R289A | AGCUUUG | >100 | n.d. | n.d. | n.d. |
| RBM39 mRRM2 | AGCUUUG | >100 | n.d. | n.d. | n.d. |
| RBM39 RRM12 | ssSL3 | 0.6 | 0.93 | −333 | 297 |
| RBM39 RRM12 | SLi | 8 ± 3 | 0.86 | −89 | 60.6 |
| Protein | Peptide | $K_d$ (µM) | n | ΔH (kcal/mol) | −TΔS (kcal/mol) |
| RBM39 RRM3 | KSRWDETP | 19 ± 5 | 0.65 | −20 | 13.6 |
| RBM39 mRRM3 | KSRWDETP | >100 | n.d. | n.d. | n.d. |

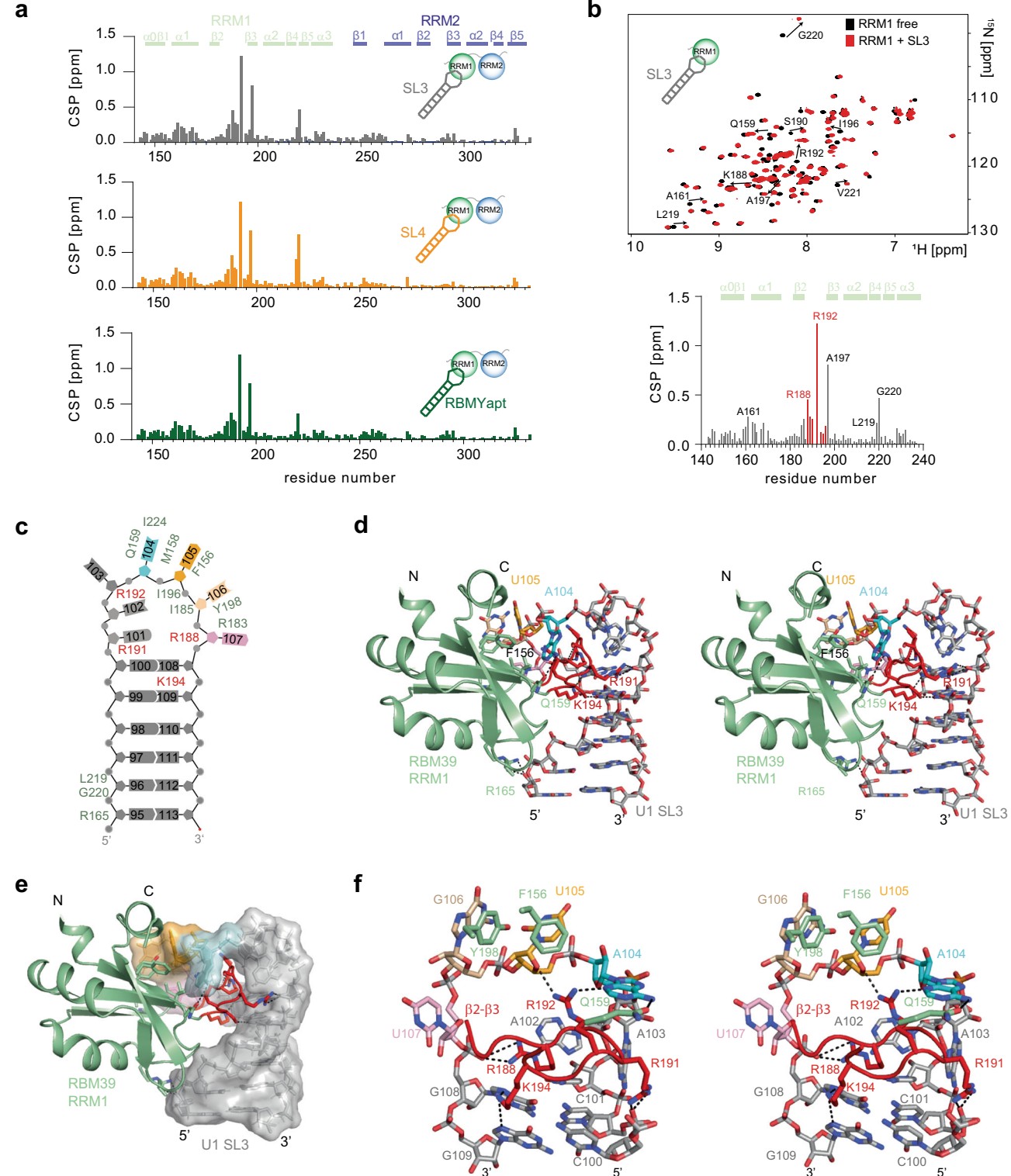

**Fig. 6 | Structural basis for RNA stem loop shape recognition by RBM39.**
**a** Barplots showing the amide chemical shift perturbation as a function of the
RBM39 RRM12 sequence observed when the protein was titrated with either
U1 snRNA SL3, SL4 or the RBMY aptamer. For each titration, the protein was con-
centrated to 200 μM and the RNA stock was at 2 mM. **b** Overlay of the 2D $^{15}$N-$^{1}$H
HSQC spectra of $^{15}$N-labelled RRM1 (200 μM) recorded before and after addition of
one equimolar amount of U1 snRNA stem loop 3 (stock solution at 2 mM). Spectra

corresponding to free and bound protein are coloured in black and red, respec-
tively. Normalised amide CSPs are plotted as a function of the sequence of RRM1.
**c** Schematic representation of the protein–RNA contacts observed in the solution
structure of RRM1 in complex with SL3. **d** Stereo view of the solution structure of
RRM1 in complex with U1 SL3. **e** Surface representation. **f** Stereo view showing the
role of the loop β2-β3 in the recognition of the stem loop shape.

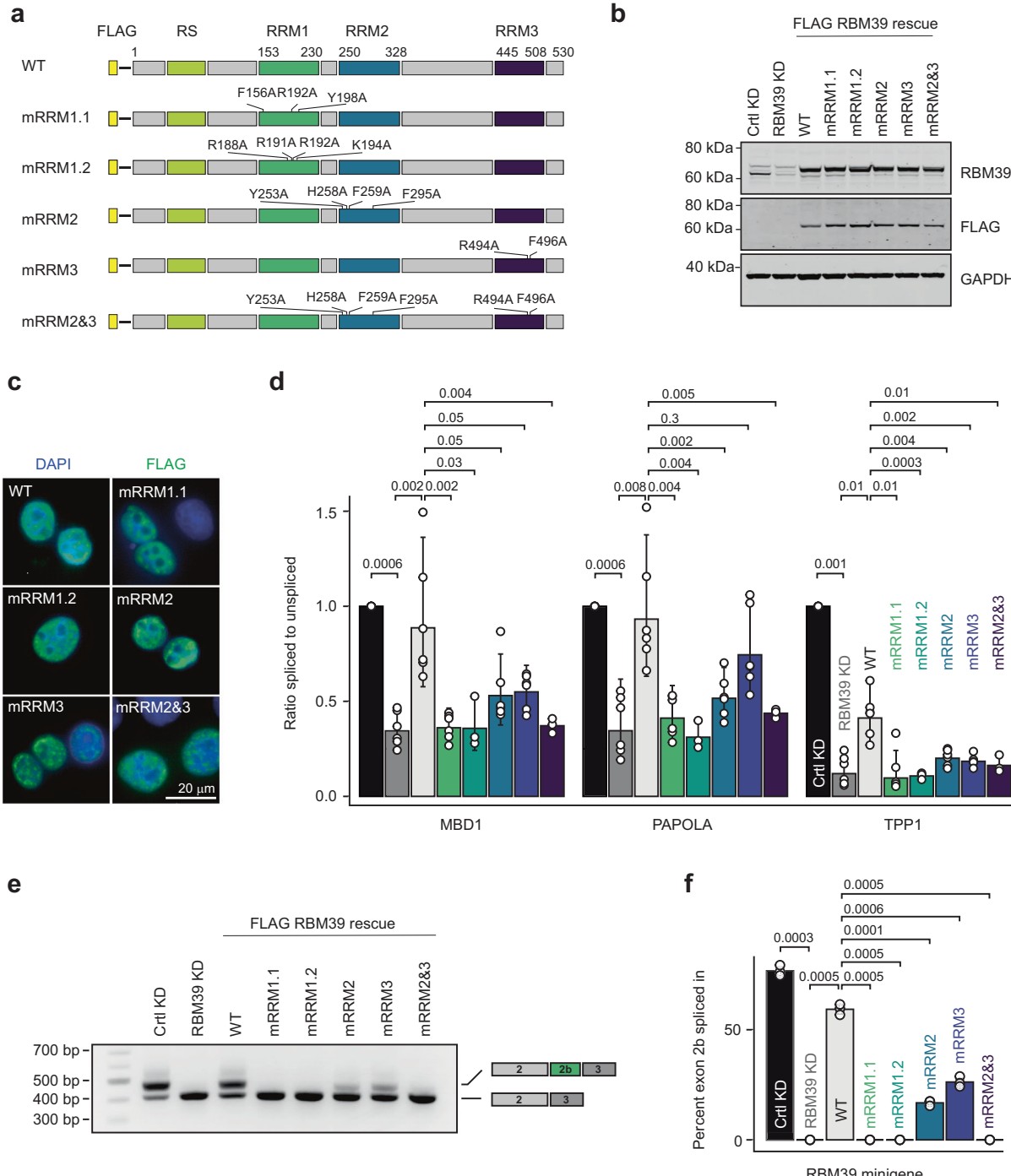

**Fig. 7 | Functional relevance of the RBM39–RNA interfaces in splicing.**
**a** Schematic representation of the RBM39 isoforms used to rescue RBM39 KD. mRRM1.1 is mutated on RRM1 β-sheet surface, mRRM1.2 is mutated on the loop β2-β3, mRRM2 is mutated on the RRM2 RNA-binding interface and mRRM3 is mutated on its U2 snRNP binding interface. **b** Western blot analysis of RBM39 levels upon control knockdown (Ctrl KD), RBM39 knockdown (RBM39 KD), and rescue with FLAG-RBM39. HeLa cell extracts were subjected to SDS–PAGE and Western blotting with anti-RBM39 and anti-FLAG antibodies. GAPDH served as a loading control. *n* = 3 or 6 **c** Immunofluorescence analysis of FLAG-RBM39 constructs upon transient expression in HeLa cells. Exogenous RBM39 was visualised using anti-FLAG

antibodies and nuclei were counterstained with DAPI (*n* = 3). **d** RT-qPCR measurements displayed as the ratio of spliced to unspliced isoform for three retained introns identified by RNA-Seq (MBD1, PAOLA and TPP1). Average values and standard deviations of three (mRRM1.2, mRRM2&3) or six (Ctrl KD, RBM39 KD, WT, mRRM1.1, mRRM2, mRRM3) biological replicates (*n* = 3 or 6) are shown. *P* values were computed from log-transformed ratios using two-sided unequal variances Welch's *t* test[65]. **e** Agarose gel showing the efficiency of poison exon inclusion as assessed by RT-PCR. **f** Barplot showing the percentage of exon 2b inclusion in the different conditions. Average values and standard deviations of three biological replicates (*n* = 3) are shown. *P* values were computed using two-sided *t* test.

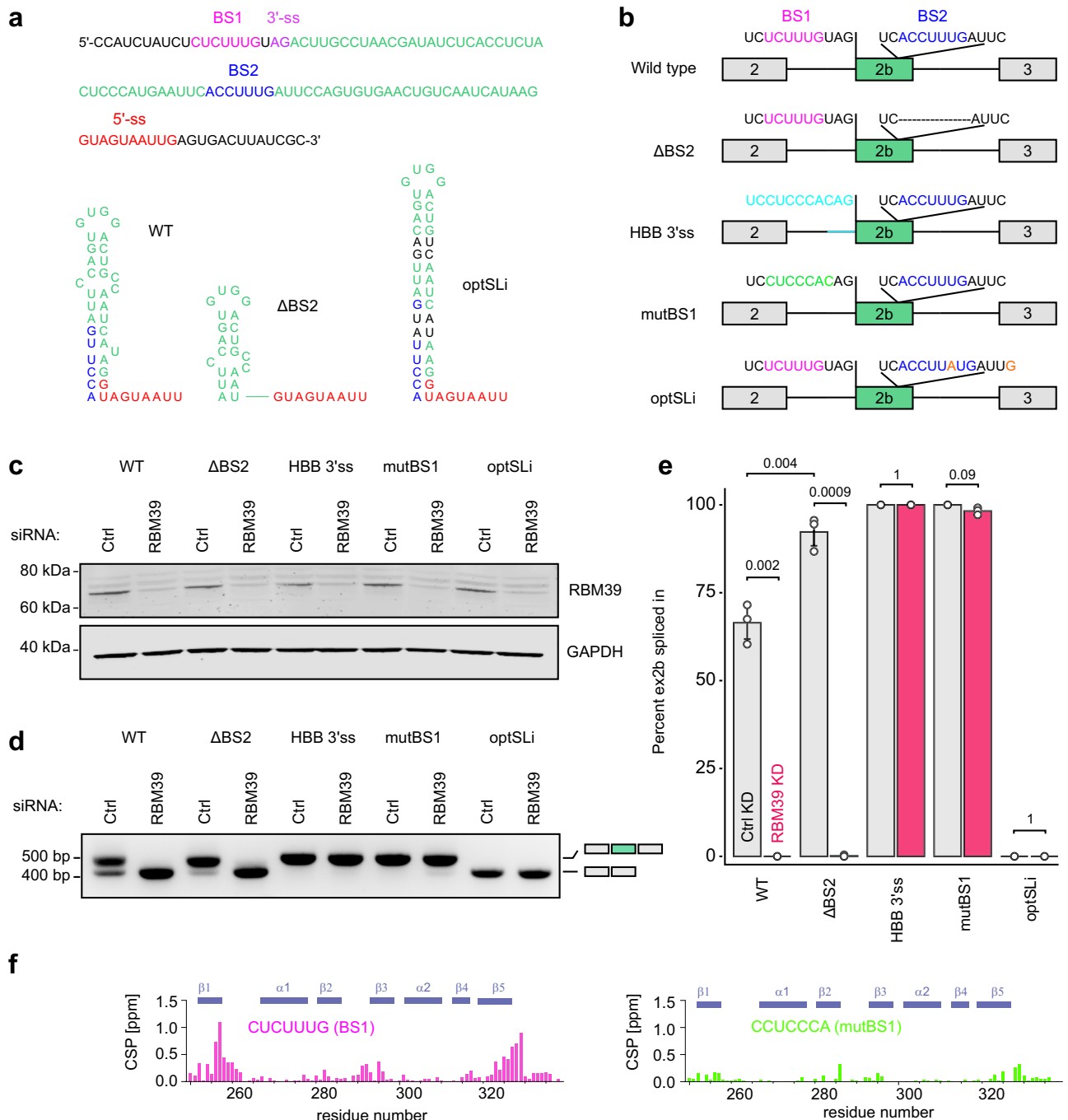

**Fig. 8 | RBM39 controls its homoeostasis through non-canonical 3′-splice site selection. a** Annotated sequence of the *RBM39* exon 2b and its flanking regions. Below, the predicted inhibitory secondary structure SLi is displayed as well as the ΔBS2 and the optSLi stem loops. **b** Schematic representations of the different minigene constructs (WT, ΔBS2, HBB 3′ss, mutBS1 and optSLi). **c** Western blot analysis of the RBM39 knock down. GAPDH served as a loading control (*n* = 3). **d** Agarose gel showing the results of the RT-PCR using the different minigene constructs (WT, ΔBS2, HBB 3′ss, mutBS1 and optSLi) in control and RBM39 KDs. **e** Barplot showing the percentage of poison exon inclusion as determined by RT-PCR for the different minigene variants upon Ctlr and RBM39 KD. Average values and standard deviations of three biological replicates (*n* = 3) are shown. *P* values were computed using two-sided *t* test. **f** Plot showing the amide chemical shift perturbations observed upon titration of RBM39 RRM2 with BS1 or mutBS1. The protein was concentrated to 200 µM and the RNA stock was at 2 mM.

spectroscopy that RBM39 RRM12 binds to SLi in vitro (Supplementary Fig. 11). Both proposed scenarios are illustrated in Fig. 9.

## Possibilities for the development of innovative anti-cancer therapy

In AML[22] and in high-risk glioblastomas cases[16], response to aryl sulfonamides treatment correlated with high expression level of DCAF15.

This observation suggests that the identification of alternative approaches triggering the degradation of RBM39 independently of DCAF15 could be beneficial for cancer treatments. By deciphering the molecular mechanisms governing the homoeostasis of RBM39, we provide important data to manipulate this negative feedback loop mechanism and to trigger the depletion of RBM39 independently of DCAF15. By pushing the splicing equilibrium towards the constitutive

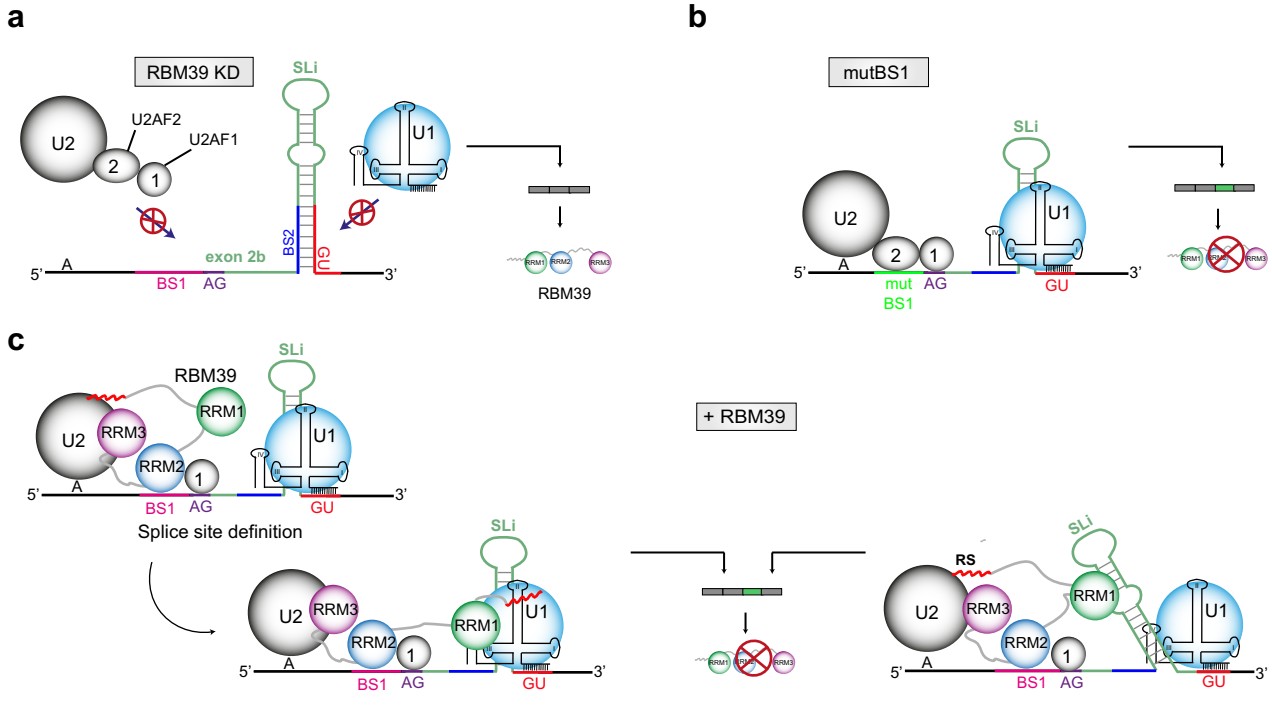

**Fig. 9 | Molecular mechanism of RBM39 autoregulation. a** Schematic representation of the RBM39 poison exon (exon 2b) and the *cis*-acting RNA elements governing its inclusion. In the absence of RBM39, the splice sites of the poison exon are not recognised efficiently. The proteins labelled with 1 and 2 correspond to U2AF1 and U2AF2, respectively. **b** When the RBM39 binding motif BS1 is mutated, U2AF2 binds to the 3′-ss and promotes poison exon inclusion independently of RBM39. **c** In the wild-type situation, RBM39 contacts the 3′-ss via RRM2 and stabilises the U2 snRNP on the branchpoint via RRM3 and the RS domain. RRM1 interacts with an RNA stem loop, which could be located either in the pre-mRNA (e.g. SLi) or in a spliceosomal RNA (e.g. U1 snRNA).

inclusion of the poison exon, the expression of RBM39 could be shut down, resulting in cancer cell death in a DCAF15-independent manner. The manipulation of the autoregulation mechanism using RNA therapeutics could represent a novel strategy to deplete RBM39 from cancer cells. A similar strategy was recently used to lower the level of *huntingtin* mRNA and develop an innovative therapeutic approach against Huntington's disease[57]. The rapid expansion of the field of RNA therapeutics correcting splicing has provided innovative therapeutic strategies for inherited diseases[42,58–63]. In the future, it could also benefit cancer therapy.

## Methods

### Cell culture

HeLa cells were grown in Dulbecco's modified Eagle's medium (DMEM) supplemented with 10% foetal calf serum (FCS), penicillin (100 IU/ml) and streptomycin (100 µg/ml) (DMEM+/+) at 37 °C and 5% $CO_2$.

### Cloning

Plasmids allowing the expression of RBM39 RRM1, RBM39 RRM2 and RBM39 RRM12 were prepared by subcloning the corresponding *E. coli* codon optimised ORF (GeneScript) into pET26bII between the NdeI-XhoI site. RRM1 was subcloned using the sc1 and sc2 oligonucleotides; RRM2 was subcloned using the sc3 and sc4 oligonucleotides and RRM12 was subcloned using the sc1 and sc4 oligonucleotides. Mutagenesis was performed by following the quick-change protocol. Plasmids allowing the expression of protein mutants (mRRM1.1, mRRM1.2, mRRM3 and RRM3) were designed into pET26b+ and purchased (Genscript). pcDNA3.1-FLAG-GSG15-RBM39 was created by cloning codon optimised RBM39 (GeneArt) into the BamHI and NotI sites of pcDNA3.1-FLAG-GSG15 (gift of Dr. Asimina Gratsou, University of Bern). To generate pcDNA3.1-FLAG-GSG15-RBM39-deltaRRM1 and deltaRRM2, the optimised coding sequences of RBM39 missing amino acids 153–230 (RRM1) or amino acids 250–330 (RRM2) were ordered from GeneArt, amplified using the primers dj217 and dj218 and inserted into the BamHI and NotI sites of pcDNA3.1-FLAG-GSG15. To create pcDNA3.1-FLAG-GSG15-RBM39-deltaRRM3, the coding region for RBM39 amino acids 1–445 was amplified from pcDNA3.1-FLAG-GSG15-RBM39 with dj217 and mdr835 and inserted into the BamHI and NotI sites of pcDNA3.1-FLAG-GSG15. To generate pcDNA3.1-FLAG-GSG15-deltaRS-RBM39-SV40NLS, the region encompassing amino acids 146–530 was amplified using dj798 and dj799 and inserted into the BamHI and NotI sites of pcDNA3.1-FLAG-GSG15. The point mutations to disrupt RNA-binding and UHM-ULM interactions were introduced by Quick-change mutagenesis of pcDNA3.1-FLAG-GSG15-RBM39. RRM1 was mutated using the primers mdr842 (Y198A), mdr843 (F156A), mdr866 (R192A), whereas RRM2 was mutated using mdr867 (H258A, F259A), mdr864 (Y253A) and mdr865 (F295A). RRM3 was mutated using mdr841 (R494A, F496A). The pd2-N1-RBM39-minigene was generated as follows: the region spanning exon 1 to 4 of the endogenous RBM39 gene was amplified from human fibroblast gDNA using the primers dj282 and dj283 and inserted into the SalI and NotI sites of pd2-EGFP-N1 (Clonetech). Subsequently, a cryptic splice site in intron 3 was destroyed by QuikChange mutagenesis using the primer mdr848. Pd2-N1-RBM39 deltaBS1 was generated by Quikchange mutagenesis using the primer dj779. To create the pd2-N1-RBM39-optSLi plasmid, the region spanning RBM39 exons 2 – 3 with the desired point mutations was ordered by gene synthesis and cloned into the EcoRV and Not1 sites of the pd2-EGFP-N1-RBM39 minigene. To generate the plasmids pd2-N1-RBM39-deltaBS2 and pd2-N1-RBM39-HBB-3′ss, the region spanning RBM39 exons 2 and 3 between the BstX1 and EcoRV sites containing the desired mutations was ordered by gene synthesis and cloned into the pd2-N1-RBM39 minigene. To create the RBD-only

FLAG-RBM39 constructs for RNA/RNP immunoprecipitation, the RNA-binding region of RBM39 was amplified from the pcDNA3.1-FLAG-GSG15-RBM39 (WT, mutRRM1 and mutRRM2) expression vectors using dj734 and dj735 and cloned into the BamHI and NotI sites pcDNA3.1-FLAG-GSG15. For the RRM1&2 double mutant, the regions encoding RRM1 and RRM2 were first amplified from pcDNA3.1-FLAG-GSG15-RBM39 (mutRRM1 and mutRRM2) using dj734/dj737 (RRM1) and dj736/dj735 (RRM2), and the resulting PCR fragments where then fused by a second round of PCR amplification with dj734/dj735 before cloning into the BamHI and NotI sites of pcDNA3.1-FLAG-GSG15. To create the pEBFP-c1-HBB minigene with Hind3 and Sal1 restriction sites in intron 2 to facilitate the subsequent introduction of alternative exons, the regions of HBB exon 1 to intron 2 and HBB intron 2 to exon 3 were amplified from fibroblast genomic DNA using the primers dj714, dj715, dj716 and dj717. After fusion of the two fragments by PCR with dj714 and dj717, the HBB gene was inserted into the Bgl2 and Xba1 sites of pEBFP-c1. To generate the pEBFP-c1-HBB-RBM39 WT and deltaBS1 minigene, the region containing RBM39 exon 2b was amplified from pd2-N1-RBM39 minigene WT or deltaBS1 using dj777 and dj778 and cloned into the Hind3 and Sal1 sites of the pEBFP-c1-HBB minigene. To create the pEBFP-c1-HBB RBM39 HBB 3'ss and mutBS1 plasmids, the region spanning RBM39 exon 2b with the desired point mutations was ordered by gene synthesis and cloned into the Hind3 and Sal1 sites of the pEBFP-c1-HBB minigene.

## Protein expression and purification

Expression of RRM1, RRM2 and RRM12 were performed in *E. coli* BL21 DE3. $^{15}$N and $^{13}$C uniform isotopic labelling was performed in M9 medium complemented by 1 g of $^{15}$NH$_4$Cl and/or 2 g of $^{13}$C glucose. ILV methyl labelling was performed in M9-D$_2$O medium in presence of $^{15}$N-labelled ammonium chloride, unlabelled glucose, 100 mg/L of alpha-ketobutyric acid (methyl-13C, 99%; 3,3-D2, 98%, Cambridge Isotope Laboratory) and 60 mg/L of alpha-ketoisovaleric acid (13C5, 98%; 3-D1, 98%, Cambridge Isotope Laboratory). All the recombinant protein expressions were performed at 37 °C during 4 h in the presence of 1 mM IPTG. Cell lysis was performed using a microfluidizer (3 cycles at 15,000 psi) in buffer A (10 mM Hepes pH 7.8, NaCl 1 M, Imidazole 10 mM, β-mercapto-ethanol 2.8 mM) in the presence of DNAse I (10 μg/ml), lysozyme (10 μg/ml) and anti-protease tablets (Roche). The cell lysate was clarified by centrifugation (30,000 *g*, 4 °C, 30 min) and loaded into an 5 ml HisTrap column (Cytiva) previously equilibrated in buffer A. Protein was eluted by a linear gradient of imidazole and dialysed at room temperature during 4 h in buffer B (10 mM Hepes pH 7.5, NaCl 0.25 M, β-mercapto-ethanol 2.8 mM) in presence of thrombin (50 units/10 mg of purified protein; Sigma). The resulting digest was loaded into an 5 ml HisTrap column previously equilibrated in buffer B and the flow through was collected, dialysed in buffer C (10 mM sodium phosphate buffer pH 7.0, NaCl 50 mM, DTT 2 mM). The sample was loaded on a 5 ml HiTrap SP FF column (Cytiva) and eluted with a linear gradient of buffer D (10 mM sodium phosphate buffer pH 7.0, NaCl 1 M, DTT 2 mM). RRM2 and RRM12 were dialysed in buffer E (10 mM sodium phosphate buffer pH 6.8, NaCl 50 mM, DTT 2 mM) and were further purified by size exclusion chromatography (S75, Cytiva) in buffer E. RRM1 was dialysed in buffer F (10 mM sodium phosphate buffer pH 5.5, NaCl 50 mM, DTT 2 mM) and was further purified by size exclusion chromatography in buffer F (S75, Cytiva). RRM3 and mRRM3 were expressed and purified as previously described[25]. All the protein mutants were prepared using similar protocols than wild-type proteins. Recombinant protein purification was monitored by SDS-PAGE.

## RNA production

The RNA U1 snRNA SL3, SL4 and RBMY aptamers were produced by T7-driven in vitro transcription using homemade T7 RNA polymerase. Transcriptions were performed using ssDNA templates (Supplementary Data 1). The transcription mixture was prepared as follows: 30 mM MgCl$_2$, 6 mM of each NTP, 4 mM GMP, 2.5 μM dsDNA template and 1.7 μM T7 RNA Polymerase in a transcription buffer containing 40 mM Tris-HCl pH 8.0, 1 mM spermidine, 0.01% Triton X-100 and 5 mM DTT. 1 U/mL pyrophosphatase was added to reduce magnesium phosphate accumulation. After 4 h, the reaction was stopped by adding 100 mM EDTA pH 8.0. The mixture was then centrifuged (4000*g*, 10 min) and filtered with 0.22 μm filter. The transcription mixture was then loaded into a preparative anion exchange column mounted on an HPLC system allowing the purification of RNA in denaturing conditions (80 °C, 6 M urea). The RNA was eluted with a gradient of salt, butanol-extracted, refolded and lyophilised. RNA production and purification were monitored by Urea-PAGE. Short RNA fragments were purchased (Horizon).

## Immunofluorescence

HeLa cells were grown to 80% confluency in 6-well plates and transfected with 300 ng pcDNA3.1-FLAG-RBM39 constructs using Lipofectamine2000 (Invitrogen). On the following day, 40,000 HeLa cells were re-seeded in 8-well chambers (Bioswisstec AG) and fixed after 24 h with 4% PFA for 30 min. After three washes with TBS, the cells were permeabilised and blocked using 1x TBS, 0.5% (v/v) Triton-X-100, 6% BSA at room temperature for 1 h. Ms anti-FLAG M2 antibodies (Sigma) were diluted 1:200 in TBS+/+ (1x TBS, 0.1% (v/v) Triton-X-100, 6% BSA) and incubated with the cells overnight at 4 °C. After 3 × 5 min washes with TBS+/+, the secondary antibody (Chicken anti-Mouse AF488, 1:500, Invitrogen) was diluted in TBS+/+ and bound to the primary antibody at 37 °C for 1.5 h followed by incubation at room temperature for 30 min. Then, the slides were washed 5 times with 1x TBS and mounted with Vectashield HardSet mounting medium containing DAPI (Vectorlabs). Images were acquired with a non-confocal fluorescence microscope (Leica, DMI6000 B) using the Leica Application Suite software (LAS-X) or with a non-confocal Eclipse Ti-2 epifluorescence microscope (Nikon) using the NIS-Elements AR software (Ver 5.01) using a 60x/1.4 NA oil immersion lens. For printing, brightness and contrast of the pictures were linearly enhanced.

## Immunoprecipitations and western blotting

120 μl Protein G Dynabeads (Life Technologies) per immunoprecipitation were washed three times with TBS supplemented with 0.05% NP-40 (IGEPAL CA-630) and incubated in 600 μl total volume of TBS-0.05% NP-40 with 15 μg of mouse anti-SmB/B' (Y12), mouse anti-U1A (SCBT, sc-101149), rabbit anti-RBM39 (Bethyl, A300-291A), mouse IgG (SCBT, sc-2025 or Jackson Immuno Research, 015-000-003), or rabbit IgG (SCBT, sc-2027 or Jackson Immuno Research, 011-000-003) head over tail for 1.5 h or supplemented with 1 mg/ml BSA overnight at 4 °C. The beads were subsequently washed three times with 1 ml TBS-0.05% NP-40 and resuspended in 600 μl TBS-0.05% NP-40 supplemented with 1 x Halt Protease Inhibitor Cocktail (Thermo Scientific) or 1x Halt Protease/Phosphatase Inhibitor Cocktail. For RNAse-free IP conditions TBS-0.05% NP-40 was supplemented with 1 U/ul RNAse inhibitor (NxGen, Lucigen or Ribolock/SUPERaseIN, Thermo Fisher). For nuclease treated samples, the immunoprecipitations were supplemented with 1.5 mM MgCl$_2$ f.c., 416.6 U/ml Cyanase (Ribosolutions Inc) and 333 μg/ml RNAse A (Sigma).

Hela nuclear extract (Ipracell, CC-01-20-50) was thawed on ice, cleared by centrifugation for 5–10 min at 5000–10,000*g*. 30 μl of cleared nuclear extract per 15 μg of antibody were added to the protein G–antibody complexes and incubated for 1.5 h head of tail at 4 °C. Subsequently, the beads were washed three times with TBS-0.1% NP-40, followed by a final wash with 5 min head over tail incubation. With the final wash, the beads were transferred to new tubes, wash buffer was removed, and the beads were resuspended in 60 μl of 2× LDS-loading buffer, boiled for 10 min at 70 °C and loaded on a 4–12% NuPAGE gel. The proteins were transferred to nitrocellulose membranes using the iBlot Gel Transfer Device (Life Technologies).

Membranes were blocked with 5% non-fat dry milk in TBS supplemented 0.1% with Tween or with SeaBlock (Life Technologies) for 1 h and incubated with the primary antibodies rabbit anti-U1C (Bethyl, A303-947A or ab192028), rabbit anti-RBM39 (HPA001591, Sigma), mouse anti-U1A (SCBT, sc-101149), mouse anti-SF3A3 (SCBT, sc-374464) overnight at 4 °C. After five washes with TBS-Tween, the membranes were incubated with donkey anti-mouse IRDye800CW (LI-COR Biosciences, 926- 32212) or donkey anti-rabbit IRDye800CW (LI-COR Biosciences, 926-32213) in TBS-Tween-Milk for 1.5 h followed by analysis with the Odyssey Infrared Imaging System (Li-Cor).

## RNA/RNP immunoprecipitation

For the RBM39 FLAG immunoprecipitation, HeLa cells were grown to 50% confluency in 300 cm$^2$ flasks and then transfected with 10 μg of the respective pcDNA3.1 expression vectors using Lipofectamine2000 according to the manufacturer's protocol. 48 h post-transfection, the cells were harvested with trypsin/EDTA, and the pellets were shock frozen in liquid nitrogen for storage at −80 °C until use. To prepare cellular extracts, the pellets were dissolved in 5 mL Gentle Hypotonic Lysis Buffer (10 mM Tris pH 7.2, 10 mM NaCl, 2 mM EDTA, 0.5% Triton-X-100) supplemented with 2x HALT protease inhibitor cocktail (Thermo Scientific, 78429) and 0.5 U/μL RiboLock Rnase inhibitor (Thermo Scientific, EO0381) and incubated on ice for 10 min. After adjusting the NaCl concentration to 150 mM, the extracts were incubated for another 5 min on ice and then cleared by centrifugation at 15,000g and 4 °C for 15 min. Per IP, 100 μL FLAG-M2 matrix (200 μL of a 50% solution) were washed once with matrix preparation solution (50 mM Tris pH 7.5, 150 mM NaCl) once with 0.2 M Glycine pH 3.5 (to remove unconjugated FLAG antibodies) and again twice in matrix preparation solution. To keep protein and RNA input fractions, 50 μl of the cleared extracts were boiled with 2x LDS loading buffer and 200 μl were transferred to 1 ml TRIzol (Invitrogen, 15596018) supplemented with 0.14 M β-mercaptoethanol (Applichem, A1108). Three times 1.5 mL of the extracts were then distributed to Eppendorf tubes containing FLAG-M2 matrix and incubated on a rotary wheel for 1.5 h at 4 °C. After washing the beads 5 times with HEPES NET-2 buffer (50 mM HEPES pH 7.3, 150 mM NaCl, 0.1% Triton-X-100), 1/5 of the beads was transferred into a separate Eppendorf tube and boiled in 2x LDS loading buffer, whereas the remaining beads were transferred to 1 ml TriZOL supplemented with 0.14 M β-mercaptoethanol. Proteins were analysed using 4–12% Bis-Tris polyacrylamide gels in MOPS buffer followed by Coomassie staining or western blot using the following antibodies: Mouse anti-FLAG M2 (Sigma, 1:10,000). RNA was isolated from TriReagent according to the manufacturer's protocol and then reverse transcribed using the high-capacity RNA-to-cDNA kit (Applied Biosystems, 4387406) and analysed by RT-qPCR.

## Isothermal titration calorimetry

ITC experiments were performed on a VP-ITC instrument (Microcal). Both partners were prepared in the NMR buffers (buffer E for RRM2 and RRM12 or buffer F for RRM1): the protein (250 μM for RRM1 and RRM2; 90 μM for RRM12) was injected into a solution of ssRNA (10 μM for SL3 and AGCUUUG; 7 μM for the composite RNA motif ssSL3) by 40 injections of 6 μl every 350 s at 35 °C (for RRM12) or at 40 °C (for RRM1 and RRM12). Raw data were integrated, normalised for the molar concentration and analysed using Origin 7.0 according to a 1:1 binding model. ITC experiments performed with the mutants RRMs (mRRM1.1, mRRM1.2, mRRM2, RRM3 and mRRM3) were performed with a PEAQ-ITC (Malvern). mRRM1.1, mRRM1.2 and mRRM2 titrations were performed in the same buffer than the wild-type proteins while the RRM3 titration were performed in Buffer G (Sodium-Phosphate 10 mM pH7.2, NaCl 50 mM, TCEP 0.5 mM). The proteins were diluted at 300 μM in the syringe and the ligand at 30 μM in the cell. Data were analysed using the manufacturer's software.

## RBM39 knockdown and RNA sequencing

HeLa cells, in 6-well plates at 80% confluency were transfected using Lipofectamine 2000 with either 120 pmol of Control siRNA (5′-AGGUAGUGUAAUCGCCUUGdTdT-3′) and 300 ng of pcDNA3.1, 60 pmol RBM 3′UTR (5′-GAGAAUUCAUCUUGAGUUAdTdT-3′), 60 pmol RBM CDS2 siRNA (5′-GGAAAGAGAUGCAAGGACAdTdT-3′) and 300 ng of pcDNA3.1, or 60 pmol RBM 3′UTR, 60 pmol RBM CDS2 siRNA and 300 ng of pcDNA3-FLAG-RBM39 using Lipofectamine 2000. 24 h post-transfection the cells were split and re-transfected the next day with 160 pmol of siRNAs and 300 ng of pcDNA3 expression constructs. The cells were expanded the next day and harvested 5 days post-transfection in Tri-Reagent for RNA extraction or RIPA Lysis and extraction buffer (Thermo Scientific) for immunoblot analysis. $1.5 \times 10^5$ cell equivalents were loaded on a 4–12% Bolt Bis-Tris protein gel and subjected to western blotting. Knockdown and rescue were analysed using rabbit anti-RBM39 (Bethyl, A300-291A, 1:10,000), mouse anti-FLAG (Sigma-Aldrich, F3165, 1:2000) and rabbit anti-Actin (Sigma-Aldrich, A5060, 1:2000). RNA from each condition in triplicates was processed at the genomics core facility of the University of Bern using the Illumina TruSeq Stranded mRNA Library Prep Kit and sequenced on an Illumina HiSeq3000 platform using in 2 × 150 bp paired-end sequencing cycles. To assess the ability of RBM39 mutants to promote pre-mRNA splicing, we rescued siRNA-mediated knockdown of endogenous RBM39 by co-transfection of 300 ng RNAi-resistant pcDNA3-FLAG-RBM39 constructs using the protocol described above. To study the autoregulation of RBM39, 100 ng pd2-N1-RBM39-minigene were included in the second transfection at day 3. UPF1 knockdown experiments were performed according to the same timeline employing either 120 pmol Ctrl siRNA or 120 pmol UPF1 siRNA (5′-GAUGCAGUUCCGCUCCAUUdTdT-3′) along with 300 ng of empty pcDNA3 vector. Western blots were probed with rabbit anti-RBM39 (Bethyl, A300-291A, 1:10,000), mouse anti-FLAG (Sigma-Aldrich, F3165, 1:2000), goat anti-UPF1 (Bethyl Laboratories, A300-038A, 1:1000) and mouse anti-TUB1A2 (Sigma Aldrich, T9028, 1:5000) to assess knockdown and rescue efficiencies. RNA isolated from TriReagent was Dnase treated using the Turbo DNA-free kit (Ambion) followed by reverse transcription using the high-capacity RNA-to-cDNA kit (Applied Biosystems). RT-PCR was carried out for 25 cycles using 2x KAPA Taq ready-mix (Kapa Biosystems) in 50 μl reactions containing 80 ng cDNA and 400 nM primers each, following the manufacturer's manual. qPCR was performed using the 2x MESA Green qPCR Master Mix Plus (Eurogentec) for SYBR assays in 50 μl reactions containing 32 ng cDNA at a primer concentration of 600 nM each, according to the manufacturer's instructions. Pipetting was performed using a Qiagility pipetting robot (Qiagen) and a Rotor-Gene 6000 Q cycler (Qiagen) was used for PCR amplification and fluorescence monitoring. Primer sequences are listed in Supplementary Data 1. Relative quantifications of transcripts were carried out using the ΔΔCt method[64] and statistical significances were assessed on non-transformed ΔΔCt values using the Welch's test[65] to account for unequal variance between the conditions.

## Transcriptomic analysis

RNA-seq data have been analysed for transcriptional and isoform-level changes. Common to both analyses is the mapping to the human genome GRCh38 by means of STAR aligner version 2.5.2a[66]. Annotation indexes were built based on Ensembl GTF files (release 84). Gene-level counts were computed with featureCounts[67] program of the Subread package (version 1.4.6). Differential gene expression analysis was carried out with default options with DESeq2 version 1.6.3[33]. The differential analyses have been carried out on two comparisons: KD vs control and rescue vs KD. A meta-analysis of p-values of the two comparisons determined the list of all significant events. Alternative exon usage was instead identified by using DEXseq[35] on BAM files. Ensembl isoforms were filtered for main transcripts as classified in the

APPRIS database[68]. Finally, intron retention was quantified with custom python scripts based on PySam package[69]. Briefly, a ratio between spliced and unspliced reads was calculated and the significant differences between controls, KD and rescue conditions were assessed using a modified DESeq2 run (https://github.com/Martombo/SpliceRatio).

### In vitro reconstitution of U1 snRNP

The preparation of the U1 snRNP components and the particle assembly was performed as previously described[43].

### NMR spectroscopy

All the NMR spectroscopy measurements were performed either at 308 K (for RRM2) or at 313 K (for RRM1 and RRM12) using Bruker AVIII 500 MHz, AVIII 600 MHz, AVIII 700 MHz and Avance 900 MHz spectrometers. The data were processed with Topspin3.2 (Bruker) and analysed with CARA[70]. Sequence-specific backbone and side chain assignments were achieved using the classical approach[71]. All NOE spectroscopy (NOESY) experiments were recorded with a mixing time of 80 ms to avoid spin diffusion. RNA assignment was performed using 2D $^1$H-$^1$H homonuclear TOCSY and NOESY for the ssRNA target and U1 SL3. U1 SL3 was also produced $^{15}$N/$^{13}$C-labelled and assigned by combining 3D HCCH-TOCSY and 3D $^1$H-$^{13}$C HSQC NOESY. Identification of intermolecular NOEs was achieved by comparing 2D F2f $^1$H-$^1$H NOESY and 2D F1fF2f $^1$H-$^1$H NOESY recorded with various mixing times ranging between 60 ms to 150 ms[72] for both protein–RNA complexes. For the complex RRM1-U1SL3, we also used 3D $^{13}$C-(F1 edited, F3 filtered) NOESY HSQC[73] using RRM1 $^{15}$N-$^{13}$C labelled and SL3 unlabelled.

### Structure calculation

To solve the structures of the RNA bound states of RRM1 and RRM2, the resonance assignments of the bound proteins were used for peak picking and automatic NOE assignment in the 3D NOESY spectra using UNIO–ATNOS–CANDID[74] in combination with structure calculations using CYANA. In addition, dihedral angle constraints were derived with TALOS+ using backbone chemical shifts as input[75] and hydrogen bonds were identified based on reduced amide exchange rates in $D_2O$. RNA intramolecular NOEs and protein–RNA intermolecular NOEs were picked manually. The resulting peak lists including intramolecular and intermolecular NOEs, the hydrogen bonds restraints and backbone dihedral angle restraints were combined to calculate the initial structures of the protein–RNA complex using the CYANA NOEASSIGN module[76]. The 50 lowest energy structures were refined in cartesian space using the SANDER module of AMBER20[77]. Analysis of the refined structures was performed using AMBER20 scripts and PROCHECK[78].

### Reporting summary

Further information on research design is available in the Nature Portfolio Reporting Summary linked to this article.

## Data availability

The RNA-seq data have been deposited into the GEO database under the accession code GSE202134. The atomic coordinates of the structure of the RRM1–SL3 complex have been deposited in the PDB under the accession code 7ZAP. The chemical shifts of the RRM1–SL3 complex were deposited in the BMRB under the accession code 34715. The atomic coordinates of the structure of the RRM2–AGCUUUG complex have been deposited in the PDB under the accession code 7Q33. The chemical shifts of the RRM2–AGCUUUG complex were deposited in the BMRB under the accession code 34673. Source data are provided with this paper.

## Code availability

The custom script used for the intron retention analysis were deposited in GitHub under https://github.com/Martombo/SpliceRatio.

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

## Acknowledgements

This work was supported by the Swiss National Science Foundation (SNSF) and the NCCR RNA and Diseases through the grant 51NF40-182880 (F.H.-T.A.). This project was sustained by the INSERM Transfert office through the grant COPOC2021 MAT-API-00785-A-01 (S.C.) and by the federal council of La Ligue contre le Cancer (convention N°AAPARN 2021.LCC/SeC to S.C.). This research and related results were also made possible through the support of the UK Dementia Research Institute [award number UK DRI-6005], which receives its funding from UK DRI Ltd, funded by the UK Medical Research Council, Alzheimer's Society and Alzheimer's Research UK (M.-D.R.), the NOMIS Foundation (M.-D.R.), and the John and Lucille van Geest foundation (M.-D.R.).

## Author contributions

S.C., D.J., M.D.R. and F.H.T.A. designed the research. S.C., M.M., F.M., K.R. and M.F. performed the NMR spectroscopy study. S.C. solved the structure of the protein/RNA complexes. S.C. and K.R. performed the ITC experiments. D.J., M.C. and M.D.R. performed the transcriptomic analysis and their validation. D.J. performed all the cell-based assays. S.C., D.J., M.D.R. and F.H.T.A. discussed the results and wrote the manuscript.

## Competing interests

The authors declare no competing interests.
