## [Peer Review File · Nature Communications]

Molecular basis of RNA-binding and autoregulation by the cancer-associated splicing factor RBM39REVIEWER COMMENTS

Reviewer #1 (Remarks to the Author):

In this manuscript, Campagne and colleagues characterize the RNA binding properties of RBM39 RRM1 and RRM2, and a potential role of these interactions, as well as the RRM3/UHM in mediating UHM/ULM interactions for splicing regulation. The authors present two NMR-derived structures of RRM1 and RRM2 bound to respective RNA ligands, that indicate different modes of RNA recognition of the two domains. The first RRM shows a preference for stem loops, lacking sequence specificity, while the second RRM recognizes single stranded oligonucleotides. They authors analyze the autoregulation of splicing of the RBM39 pre-mRNA. It is proposed that RBM39 promotes the inclusion of a poison exon in its own mRNAs, thereby promoting their degradation through nonsense-mediated decay. Finally, the authors propose a mechanism for the splicing and negative feedback regulation by RBM39.

The manuscript is technically sound and clear and reports interesting biochemical and structural data. However, there are several statements and conclusions that are not supported by the results and experiments provided. The model put forward sounds attractive, i.e. a division of labor by the three RRMs binding to U1, pre-mRNA and U2, but is oversimplified and not supported by the data. In my view, the authors must address these points by addition of new experimental data and a major revision before considering the manuscript for publication

Specific comments:

Major points

RBM39 role in bridging U1-U2 snRNPs

The authors claim with the title of their manuscript that RBM39 bridges the pre-mRNA, U1 and U2 snRNPs, but the evidence that they show for U1snRNP interaction is not sufficient. The authors are aware of this limitation, and at several occasions they state that the interaction they observe between the RRM1 of RBM39 and the SL3 of U1snRNA "could in principle" explain a role in bridging but experimental evidence is lacking.

The authors show that U1A/U1C and RBM39 can copurify together in CoIP experiments in an RNA dependent manner, but these data do not prove a direct interaction between the U1snRNP and RBM39. RBM39 could be present due to its binding to U2snRNP/U2AF which can be copurified with U1snRNP. In fact, the level of detection of RBM39 when pulling from U1snRNP proteins or vice versa is lower than in the case of U2snRNP-RBM39 copurification (Figure 3a/b).

The authors perform NMR titrations of U1snRNP or SL3-U1snRNA with RBM39 RRM12 constructs and see similar spectral changes. They assume that this relates to the RRM1 domain and the recognition of the SL3 of U1snRNA. However, based on the NMR structure of the complex presented, the authors

explain that the RRM recognizes the shape of the stem loop and shows no sequence specificity. In fact, in Extended Figure 4 g, an NMR titration comparison of RRM12 bound to SL3-U1snRNA, SL4-U1snRNA and another stem loop (RBM39) not related to U1snRNP are shown. These titration experiments look quite similar and the authors explain this as evidence that RBM39 presents poor/no selectivity towards the SL3-U1snRNA compared to the others. This point is contradicting the conclusions made in the main text, but the authors do not discuss this issue. Also, to clarify potential differences in binding to diverse RNA ligands binding affinities should be compared.

There is a question and possibility that RRM1 actually may bind to structured elements within the mRNA sequences adjacent to the high affinity RRM2 binding site. This is quickly brought up in the discussion, when pointing out that the simultaneous binding of RRM1 and RRM2 to the same RNA molecule highly increases the overall affinity – as they show with the chimeric RNA (SL3-U1snRNP + RRM2 high affinity sequence). This extra increase in the affinity would not occur if RBM39 interacts with two different RNA molecules (pre-mRNA and U1snRNA).

Given these points the data do not support the main conclusions made even in the title of the paper.

Mechanism of RBM39 autoregulation of its own mRNA levels

- The authors identify in their KD experiments that RBM39 autoregulates its own mRNA splicing by promoting the inclusion of a poison exon which leads to a decay on the mRNA levels. In the last part of the manuscript, they transfer the sequence of this alternative regulated exon and its adjacent regions to a minigene reporter and show that the inclusion of this exon is still influenced by the presence of RBM39. After the incorporation of modifications in the exon sequence and surroundings they observe that the dependency of RBM39 is lost for all of them (only the delta BS2 shows small dependency). Surprisingly, however, the authors show that the incorporation of the exon on the minigene in the control situation is 100%, while in the case of the native mRNA of RBM39 is below 20%. They explain this difference with the potentially higher stability of the minigene mRNA against the nonsense-mediated decay. This would explain an increase on the % of exon inclusion, but would not explain why the exon skipping mRNA is not present any more in the minigene. The fact that there is not any exon skipping suggests that other different type of regulation must be on place, obscuring the effect of RBM39.

- Finally, the authors explain the regulation process by assuming that the RRM2 of RBM39 would bind the BS1 sequence, the RRM1 would recruit the U1snRNP and the RRM3 would interact with U2snRNP. Other explanations are plausible with the presented data, for example, RRM1 could interact directly with the SLI of the exon and RRM2 interact with either BS1 or BS2, increasing the affinity for the mRNA. U1snRNP recruitment could take place via its RS domain, as have been shown before. In addition, the authors have not considered that the mutation on the BS1 region (an others), could potentially affect the recruitment of U2AF2 or other factors and therefore alter the RBM39 dependency. Thus, the interpretation and model put forward is not clearly and unambiguously supported by the experiments shown. This would need additional experimental support.

- The authors show that point mutations in the three RRM domains affect complementation upon KD of wildtype proteins in intron retention and autoregulation of splicing. However, evidence is missing to show that these mutants indeed affect RNA binding and that they are structurally intact, considering that multiple residues are mutated simultaneously. It is important to show a correlation of RNA binding (and UHM/ULM interaction for RRM3) with the effects observed in splicing to support the conclusions drawn.

- What are the binding stoichiometries of RRM2 to the various single-stranded RNAs based on the ITC data? The curve does not fully reach saturation. ITC data should be presented in a Table, listing at least K_d, binding enthalpies and stoichiometries. How do the authors rationalize the very large binding enthalpy observed for the ssSL3 RNA in Ext data fig 3, which is much more than the combination of ssRNA and SL interactions?

- Micromolar affinity for the interaction of RRM2 with ssRNA does not seem sufficient to mediate specific pre-mRNA binding in a cellular context.

Specific points

- P.7 middle paragraph: The authors should state which point mutations were used and provide evidence that these indeed affect direct RNA binding w/o disrupting the fold of the RRM domains. Are these the mutations that are discussed later only i.e. on p. 10??

- P.7, the authors note that deletion of RS and RRM3 affects specific mRNAs, suggesting a role for substrate specificity, - how is this rationalized?

- Please correct "KD" to K_d" (dissociation constant) in Fig. 3d

- Extended data figure 4 and 6: please state concentrations of proteins and RNAs employed in the NMR experiments, here and also in other figures showing NMR data.

Reviewer #2 (Remarks to the Author):

This manuscript reports a detailed mechanistic characterization of the splicing factor RBM39 by using a combination of methods including transcriptomics as well as structural insights followed by biochemical validation and cellular tests, and all this is the main strength of this paper. In particular, authors focused on the RBM39 three RNA Recognition Motifs to show that the first RRM1 binds the U1 snRNA at a loop region, the second RRM2 binds the single-stranded RNA target, and the third RRM3 binds U2/U2AF at the 3' splice site, and hence RBM39 bridges together these three components to induce splicing. In addition, the splicing events altered upon RBM39 knockdown revealed a poison exon within the RBM39

transcript which is used for the auto-regulation of this factor, with relevance to cancer. Overall, this paper is great because it focuses on important mechanistic aspects using a variety of techniques while the NMR data provides an atomic-resolution map of key interactions between RBM39 and binding partners, which is very valuable. Hence, this manuscript should be acceptable for publication in Nature Communications, if authors can satisfy the minor comments below.

1. The 'applied' implications of this research on cancer therapeutics are important, however the nature of the data is actually mechanistic and more basic than applied. Hence, the Abstract excessively focuses on the existing drugs targeting RBM39 in cancers where this factor acts as an oncogene, and in the dependence of these drugs on DCAF15. I find this excessive because nowhere on the paper was this drug tested, and none of the experiments involved DCAF15. I recommend an extensive rewriting of the Abstract to focus on the main and very valuable mechanistic insights of this work, rather on the implications for drug design. This work is great on its own, without having to invoke too much the drug aspects.

2. The transcriptomic data analysis is a bit limited, as more insights could be derived from the RBM39 depleted cells. Can we compare the RBM39 regulated events (changed in knockout and rescued by the full-length) to other transcriptomes, or to cancer samples? Just mine a bit more the existing data.

3. In particular, one is kept wondering whether the RBM39 poison exon is found more included in RBM39 overexpressing cancers. Does this exon trigger NMD in actual tumor samples, considering that NMD may be repressed in them? The TCGA data should shed light on this.

4. While this work focuses on the RRM, authors wasted a great opportunity to test the N-terminal RS domain. Can the RS-domain deficient mutants rescue the RBM39 functions?

5. At the end of page 4, authors mention that the rescue experiment ensures that the splicing targets elucidated by knockdown are direct. While this rescue is key to rule out off-target effects of the knockdown, it is still possible that the rescue RBM39 regulates splicing factor(s) that in turn regulate the splicing event. Instead, the direct splicing targets are supported by experiments showing direct binding of protein to the RNA in cells, and/or by in vitro splicing/binding experiments with purified components. Please rephrase.

6. Figures 1F and 2E, these RT-PCR experiments should be done in experimental triplicates with means and standard deviations like in Figure 7B.

7. Figure 7B can be a bit misleading, because the mutant minigenes are not expected to affect the endogenous RBM39 splicing pattern, make sure this is clear.

Reviewer #3 (Remarks to the Author):

In this manuscript, "The cancer-associated RBM39 bridges the pre-mRNA, U1 and U2 snRNPs to regulate alternative splicing", Campagne et al. used functional and structural approaches to understand the role of RBM39 in RNA metabolism. They demonstrated that RBM39 actively participates in splice site selection and autoregulates through a negative feedback loop mechanism by controlling the inclusion of the poison exon in its own pre-mRNA. All three RNA recognition motifs (RRM1/2/3) are essential for contacting the splicing machinery at both splice sites as well as its pre-mRNA in order to achieve the autoregulation. The authors provide important data to manipulate this negative feedback loop

mechanism in order to trigger the depletion of RBM39 independently of DCAF15.

Thus, this manuscript might provide new therapeutic strategies for patients with leukemias and lymphomas that express high levels of DCAF15. However, this manuscript needs to be improved as follows:

Major concern:

In the results section describing Figure 7, the authors write, “Compared to the endogenous RBM39 mRNA, poison exon inclusion was more efficient in the minigene, likely because the minigene mRNA is less susceptible to nonsense-mediated decay”. First please add a reference to support this claim. If indeed the HBB minigene is sensitive to the NMD pathway you need to add an additional mini-gene that is not sensitive to NMD, and show how RBM39 influences the inclusion level of the poison exon in this mini-gene. As the HBB minigene does not mimic the genomic environment of the RBM39 gene, and the minigene is not under the regulatory system of the endogenous promoter, all results obtained from this minigene need to be strengthened. For example, as elongation rate influences inclusion of exons, promoters can modulate exon inclusion. Therefore, it is important to show the interplay between RBM39 and the inclusion level of the poised exon using two different promoters (a strong and a weak one). Further, RNA secondary structure can be influenced by the elongating polymerase. To confirm that the inclusion of poison exon is independent of RBM39 and controlled by an inhibitory stem loop, the authors should choose another two minigenes and insert the poison exon and approximately 100 nucleotides of its flanking intronic region into these minigenes and evaluate splicing.

Minor concerns:

The authors write, “Furthermore, the RS domains of RBM39 and U2AF2 were shown to promote liquid–liquid phase separation and to favor 3′ splice site recognition”. It is unclear how this sentence is relevant to the rest of the manuscript.

Figure 1C: The functional contribution of each RRM to the three intron retention events detected by DEXseq. Schematics of these genes and the exact intron retention events that were examined should be shown so that it is clear what events the authors are referring to.

Figure 2C: “All the constructs were moderately expressed and none of the deletions affected RBM39 subcellular localization”. Please add a graph showing a quantification to support this sentence.

Extended figure 2A: It is unclear which intron retention events were used for the validation of the RNA-seq data shown for TPP1, MBD1, and PAPOLA. Give a number for each intron or exon and relate it to UCSC genome browser.

Figure 3A: Figure 7d illustrates a model for poison exon inclusion within the context of RBM39 autoregulation, showing that the RBM39 domain interacts with U1 (via RRM1) and with U2 and U2AF65 (via RRM3) and that RBM39 interacts with U1, U2, and U2AF65 form a complex that is crucial for exon 2b inclusion level. Therefore, please add western blotting showing that U2AF65 co-precipitates with all snRNPs.

REVIEWER COMMENTS

Reviewer #1 (Remarks to the Author):

In this manuscript, Campagne and colleagues characterize the RNA binding properties of RBM39 RRM1 and RRM2, and a potential role of these interactions, as well as the RRM3/UHM in mediating UHM/ULM interactions for splicing regulation. The authors present two NMR-derived structures of RRM1 and RRM2 bound to respective RNA ligands, that indicate different modes of RNA recognition of the two domains. The first RRM shows a preference for stem loops, lacking sequence specificity, while the second RRM recognizes single stranded oligonucleotides. The authors analyze the autoregulation of splicing of the RBM39 pre-mRNA. It is proposed that RBM39 promotes the inclusion of a poison exon in its own mRNAs, thereby promoting their degradation through nonsense-mediated decay. Finally, the authors propose a mechanism for the splicing and negative feedback regulation by RBM39.

The manuscript is technically sound and clear and reports interesting biochemical and structural data. However, there are several statements and conclusions that are not supported by the results and experiments provided. The model put forward sounds attractive, i.e. a division of labor by the three RRMs binding to U1, pre-mRNA and U2, but is oversimplified and not supported by the data. In my view, the authors must address these points by addition of new experimental data and a major revision before considering the manuscript for publication.

Specific comments:

Major points

1-RBM39 role in bridging U1-U2 snRNPs

The authors claim with the title of their manuscript that RBM39 bridges the pre-mRNA, U1 and U2 snRNPs, but the evidence that they show for U1snRNP interaction is not sufficient. The authors are aware of this limitation, and at several occasions they state that the interaction they observe between the RRM1 of RBM39 and the SL3 of U1snRNA "could in principle" explain a role in bridging but experimental evidence is lacking.

The reviewer comment is justified.

Accordingly, we modified the title of the manuscript. The novel title is: "Molecular basis of RNA-binding and autoregulation by the cancer-associated splicing factor RBM39". In addition, we removed the paragraph of the discussion entitled: "RBM39 connects U1, U2 snRNPs and the pre-mRNA".

The authors show that U1A/U1C and RBM39 can copurify together in CoIP experiments in an RNA dependent manner, but these data do not prove a direct interaction between the U1snRNP and RBM39. RBM39 could be present due to its binding to U2snRNP/U2AF which can be copurified with U1snRNP. In fact, the level of detection of RBM39 when pulling from U1snRNP proteins or vice versa is lower than in the case of U2snRNP-RBM39 copurification (Figure 3a/b).

Using co-immunoprecipitation experiments in nuclear extracts, we show that RBM39 interacts with U1 and U2 snRNP. Further, we show that the interaction with U2 snRNP is RNA independent while RNase treatment impaired the interaction with U1 snRNP. We agree with the reviewer that the observed co-precipitation does not mean that the interactions are direct. In particular, the reviewer points to a lower level of detection of RBM39 when pulling down U1 snRNP compared to when pulling down U2 snRNP. To directly compare the amount of co-precipitating U1 and U2 snRNP, we repeated the RBM39 IPs from nuclear extract and migrated the gel longer to get a clean separation between the U1-A band (34 kDa) and a strong unspecific band (20 kDa), which was present in the original figure and may have affected signal quantification. The results are presented below and have been included in figure 4c:

The new results are cleaner, but the conclusion remains the same. Compared to the input, the amount of the U2 snRNP protein SF3A3 in the IP fraction is indeed higher than the U1 snRNP protein U1-A. This might reflect a higher affinity interaction between RBM39 and U2 snRNP compared to U1 snRNP, or as suggested by the reviewer, indicate that the interaction with U1 snRNP is indirect.

Thus, we decided to further study the interaction between RBM39 and the splicing machinery in HeLa cells transfected with plasmids expressing FLAG-tagged wild-type RBM39 or mutants lacking the RS domain, the RRM1, the RRM2 or the RRM3. FLAG-IPs were performed under conditions that previously allowed us to study the interaction between the splicing factor FUS and U1 snRNP (Jutzi, Campagne, *et al.*, Nat. Comm. 2020). In contrast to what we observed in HeLa nuclear extracts, none of the FLAG-RBM39 proteins was able to pull down the U1 snRNP factor U1-C, while all constructs except Δ RS and Δ RRM3 efficiently pulled down SF3A3, the U2 snRNP component. In summary, we concluded that the interaction between U1 snRNP and RBM39 is weak, RNA-dependent and may indeed occur indirectly in assembled pre-spliceosome complexes while the interaction with U2 snRNP is direct and mediated by the RRM3 and the RS domain of RBM39.

Those results have been included in Figure 4. Panel 4d shows the FLAG-IP results while panel 4e shows the quantification of the ratio SF3A3/FLAG.

The authors perform NMR titrations of U1snRNP or SL3-U1snRNA with RBM39 RRM12 constructs and see similar spectral changes. They assume that this relates to the RRM1 domain and the recognition of the SL3 of U1snRNA. However, based on the NMR structure of the complex presented, the authors explain that the RRM recognizes the shape of the stem loop and shows no sequence specificity. In fact, in Extended Figure 4 g, an NMR titration comparison of RRM12 bound to SL3-U1snRNA, SL4-U1snRNA and another stem loop (RBM Yapt) not related to U1snRNP are shown. These titration experiments look quite similar and the authors explain this as evidence that RBM39 presents poor/no selectivity towards the SL3-U1snRNA compared to the others. This point is contradicting the conclusions made in the main text, but the authors do not discuss this issue. Also, to clarify potential differences in binding to diverse RNA ligands binding affinities should be compared.

We think the reviewer comment is justified and therefore adapted our interpretation.

In the NMR titrations of RBM39 RRM12 by U1 snRNP, SL3-U1snRNA and SL4-U1snRNA we observed similar chemical shift perturbations of the RBM39 ILV methyl resonances (these titrations are now presented in Extended data Fig. 6). As proposed by the reviewer, these experiments suggest that RBM39 RRM12 interacts with RNA stem loops in a poorly specific manner. We therefore discarded all aspects regarding any specificity for U1 snRNA SL3.

To strengthen this point, we further analyzed the NMR titrations performed using ¹⁵N-labeled RBM39 RRM12 and *in vitro* transcribed SL3, SL4 and the RBMY aptamer. When plotting the chemical shift perturbations against the sequence of RRM12, we observed large CSP in the RRM1 sequence while the RRM2 resonances were poorly perturbed. These results have been included in Figure 6a:

Finally, as suggested by the reviewer, we used ITC to quantify the strength of the interactions between RBM39 RRM1 and SL3 or SL4. We determined that RBM39 RRM1 binds SL3 and SL4 with dissociation constants of 15 ± 3 and 11 ± 4 μ M, respectively. The results are now included in Table 1.

There is a question and possibility that RRM1 actually may bind to structured elements within the mRNA sequences adjacent to the high affinity RRM2 binding site. This is quickly brought up in the discussion, when pointing out that the simultaneous binding of RRM1 and RRM2 to the same RNA molecule highly increases the overall affinity – as they show with the chimeric RNA (SL3-U1snRNP + RRM2 high affinity sequence). This extra increase in the affinity would not occur if RBM39 interacts with two different RNA molecules (pre-mRNA and U1snRNA).

There is indeed a possibility that RBM39 recognizes high-affinity motifs in the sequence of mRNAs using RRM1 and RRM2. However, the RNA immunoprecipitation experiments performed using FLAG-RBM39 argue that RRM2 is the major interface with mRNA, since RRM2 mutations virtually abolish the ability of RBM39 to interact with mRNPs (Fig. 5). Nevertheless, to investigate the idea of bipartite RNA-binding in the context of RBM39 autoregulation, we investigated the binding of RBM39 RRM12 on the inhibitory stem loop SLi (which includes the potential RRM2 binding site BS2) using NMR spectroscopy and ITC.

Using ITC (see figure below a), we determined a dissociation constant between RBM39 RRM12 and SLi of $8 \pm 3 \mu\text{M}$ ($n=0.83$; $\Delta H = -89 \text{ kcal/mol}$; $-\Delta\Delta S = 60.6$). The ITC experiment suggests that RBM39 RRM12 binds SLi slightly better than SL3 ($K_d = 15 \mu\text{M}$) but with a considerably lower affinity than ssSL3 ($K_d = 0.66 \mu\text{M}$). Our additional experiments therefore indicate that SLi and BS2 are not a good target for the binding of both RRM1 and RRM2 of RBM39. This is in perfect agreement with our functional data.

To further confirm this conclusion, we investigated the interaction using NMR spectroscopy. First, we performed a titration of ^{13}C -ILV-labeled RBM39 RRM12 with SLi (see below b). We observed similar chemical shift changes on the methyl resonances than when the protein was titrated with SL3 or SL4, in agreement with our ITC results (see figure below, panel b). Second, we tested a similar titration using ^{15}N -labeled RBM39 RRM12 with SLi (blue spectra). We observed that most of the RRM1 amide resonance shifts as when the protein is titrated by SL3 (red spectra) while the resonances of RRM2 slightly shift but not as when the protein is titrated with AGCUUUG (yellow spectra, see below c). In this titration, we observed that many amide signals experienced strong line broadening and vanished beyond detection. We interpret this line broadening effect as a sign of chemical exchange: BS2 could be bound by RRM2 or form a double stranded region with the 5'-splice site (see below d). These results have been included in Extended data Fig. 11.

Given these points the data do not support the main conclusions made even in the title of the paper.

We would like to thank the reviewer 1 for his constructive feedback that motivated us to investigate the interaction between U1 snRNP and RBM39 in more details. While we had evidence from *in vitro* NMR spectroscopy that this interaction could be direct, we could now demonstrate that the interaction is weak, may be indirect in the cell and that RBM39 is able to bind diverse RNA stem loops in a poorly specific manner. Such stem loops could be present either in the pre-mRNA or in spliceosomal U snRNPs. Accordingly, we modified the manuscript and strongly toned down the role of RBM39 as a potential linker between U1 and U2. Some potential mechanisms of action of RBM39 during splicing are now illustrated in a new figure 9.

2- Mechanism of RBM39 autoregulation of its own mRNA levels

- The authors identify in their KD experiments that RBM39 autoregulates its own mRNA splicing by promoting the inclusion of a poison exon which leads to a decay on the mRNA levels. In the last part of the manuscript, they transfer the sequence of this alternative regulated exon and its adjacent regions to a minigene reporter and show that the inclusion of this exon is still influenced by the presence of RBM39. After the incorporation of modifications in the exon sequence and surroundings they observe that the dependency of RBM39 is lost for all of them (only the delta BS2 shows small dependency). Surprisingly, however, the authors show that the incorporation of the exon on the minigene in the control situation is 100%, while in the case of the native mRNA of RBM39 is below 20%. They explain this difference with the potentially higher stability of the minigene mRNA against the nonsense-mediated decay. This would explain an increase on the % of exon inclusion, but would not explain why the exon skipping mRNA is not present any more in the minigene. The fact that there is not any exon skipping suggests that other different type of regulation must be on place, obscuring the effect of RBM39.

This point was raised by several reviewers, showing that the artificial nature of the HBB minigene interfered with an unequivocal interpretation of the results. To avoid problems associated with the heterologous context of the poison exon, we repeated the experiments using a minigene encompassing RBM39 exons 1 to 4 (with full-length introns between exons). In this genetic context, we observe both poison exon inclusion and exclusion in the control condition, which confirms that the alternative splicing regulation is intact in this minigene. Moreover, the inclusion efficiency of the poison exon in the control situation is about 66%, which accurately mirrors the inclusion efficiency in the endogenous mRNA under NMD-compromised conditions. Using this improved minigene system, we could confirm the conclusions drawn from previous experiments in the context of the β -globin minigenes. The splicing of the poison exon is determined by two *cis*-acting RNA elements:

- A *cis*-RNA element RBM39-independent: the inhibitory stem loop SLi which sequesters the 5'-splice site;
- A *cis*-RNA element RBM39-dependent located upstream the 3'-splice site that we called in the manuscript BS1.

The new results are now included as part of Fig. 8:

Even if poison exon inclusion is favored in the context of the β -globin minigene, the principles of RBM39-dependent poison exon inclusion are preserved, which adds additional support to our conclusions. The results obtained in the context of the β -globin minigene are now displayed in Extended data Fig. 10.

- Finally, the authors explain the regulation process by assuming that the RRM2 of RBM39 would bind the BS1 sequence, the RRM1 would recruit the U1snRNP and the RRM3 would interact with U2snRNP. Other explanations are plausible with the presented data, for example, RRM1 could interact directly with the SLi of the exon and RRM2 interact with either BS1 or BS2, increasing the affinity for the mRNA. U1snRNP recruitment could take place via its RS domain, as have been shown before. In addition, the authors have not considered that the mutation on the BS1 region (an others), could potentially affect the recruitment of U2AF2 or other factors and therefore alter the RBM39 dependency. Thus, the interpretation and model put forward is not clearly and unambiguously supported by the experiments shown. This would need additional experimental support.

We took the remark raised by the reviewer into consideration and now present two possibilities for the final model:

- (i) RBM39 helps U2 snRNP binding to the branch point of the poison exon. RRM3 and the RS domain participate in the U2 snRNP recruitment, RRM2 selects the 3'-splice site. Once U2 is bound upstream the exon, RBM39 RRM1 could associate with U1 snRNP and help the formation of complex A.
- (ii) A similar situation but with RRM1 binding to SLi.

However, our data rule out a direct interaction between RRM2 and BS2 since in Δ BS2 mutant poison exon inclusion is abolished by RBM39 knock-down.

Here is our revised model (Fig. 9):

- The authors show that point mutations in the three RRMs affect complementation upon KD of wildtype proteins in intron retention and autoregulation of splicing. However, evidence is missing to show that these mutants indeed affect RNA binding and that they are structurally intact, considering that multiple residues are mutated simultaneously. It is important to show a correlation of RNA binding (and UHM/ULM interaction for RRM3) with the effects observed in splicing to support the conclusions drawn.

We agree with the reviewer that evidence confirming structural integrity of the mutated RRM domains and loss of their respective binding partners is an important control to validate the conclusions from the functional experiments.

In order to correlate the loss of function of RBM39 mutants with perturbed RNA-binding activity or ULM binding activity, we subcloned the different RRM mutants and also the wt RRM3 in pET26b and produced recombinant proteins in *E. coli*. The proteins were purified, and the folding was verified using NMR spectroscopy. While mRRM1.1, mRRM1.2 and mRRM2 were expressed in rich medium (without isotope labeling), RRM3 and mRRM3 were produced ¹⁵N-labeled in M9 minimum medium. The NMR spectra of the proteins demonstrate that the mutations do not disturb the fold of the RRMs.

We then used ITC to monitor the binding of mRRM1.1, mRRM1.2 and mRRM2 to the U1 snRNA stem loop 3 or to the ssRNA motif AGCUUUG and confirm that all mutations strongly impair RNA-binding.

For the UHM-ULM interactions, we combined NMR spectroscopy & ITC. Using NMR spectroscopy, we observed large amide chemical shift perturbations upon addition of the SF3b155 ULM5 (sequence: KSRWDETP) to the RRM3 but not to the mRRM3, suggesting that mRRM3 lost its ULM binding activity. We could confirm this observation using ITC since RRM3 binds to the SF3b155 ULM5 with a dissociation constant of $19 \pm 5 \mu\text{M}$ while we were not able to determine a dissociation constant for mRRM3 binding to the peptide. Our experimental data shows that mutations in RRM3 impaired the ULM binding activity without disturbing the RRM fold.

The results are now described in Extended data Fig. 9 and are presented below:

a**b****c****d****e**
- What are the binding stoichiometries of RRM2 to the various single-stranded RNAs based on the ITC data? The curve does not fully reach saturation. ITC data should be presented in a Table, listing at least Kd, binding enthalpies and stoichiometries. How do the authors rationalize the very large binding enthalpy observed for the ssSL3 RNA in Ext data fig 3, which is much more than the combination of ssRNA and SL interactions?

The binding stoichiometry, the Kd and binding enthalpies determined by ITC, where possible, are now listed in Table 1. We are currently further investigating the binding of the double domain constructs on different RNA targets and investigating potential differences with isolated domains. These structural investigations could explain why there is a difference between the binding enthalpy observed for the ssSL3 RNA compared to the combination of ssRNA and SL interactions.

- Micromolar affinity for the interaction of RRM2 with ssRNA does not seem sufficient to mediate specific pre-mRNA binding in a cellular context.

We showed previously the functional relevance of micromolar affinity in splicing regulation (see work on SRSF1, tra2beta, hnRNP A1, etc ...). More recently, we showed that FUS binding to RNA with micromolar affinity also has effects on RNA processing (Jutzi D*, Campagne S*, *et al.*, Nature Comm. 2020, doi: 10.1038/s41467-020-20191-3).

Specific points

- P.7 middle paragraph: The authors should state which point mutations were used and provide evidence that these indeed affect direct RNA binding w/o disrupting the fold of the RRM domains. Are these the mutations that are discussed later only i.e. on p. 10??

This has been clarified in the main text. The folding of the different mutants as well as their RNA-binding activities have been assessed (see above).

- P.7, the authors note that deletion of RS and RRM3 affects specific mRNAs, suggesting a role for substrate specificity, - how is this rationalized?

The role of the RS domain has been investigated in further detail. We found that RS deletion impairs the recruitment of RBM39 to nuclear speckles and the results are summarized in a novel figure 3. Nevertheless, the sentence is speculative, and we therefore removed it.

- Please correct "KD" to Kd" (dissociation constant) in Fig. 3d

This has been changed according to the reviewer suggestion.

- Extended data figure 4 and 6: please state concentrations of proteins and RNAs employed in the NMR experiments, here and also in other figures showing NMR data.

Protein and RNA concentrations employed in the NMR experiments have been stated in the associated figure legends.

Reviewer #2 (Remarks to the Author):

This manuscript reports a detailed mechanistic characterization of the splicing factor RBM39 by using a combination of methods including transcriptomics as well as structural insights followed by biochemical validation and cellular tests, and all this is the main strength of this paper. In particular, authors focused on the RBM39 three RNA Recognition Motifs to show that the first RRM1 binds the U1 snRNA at a loop region, the second RRM2 binds the single-stranded RNA target, and the third RRM3 binds U2/U2AF at the 3' splice site, and hence RBM39 bridges together these three components to induce splicing. In addition, the splicing events altered upon RBM39 knockdown revealed a poison

exon within the RBM39 transcript which is used for the auto-regulation of this factor, with relevance to cancer. Overall, this paper is great because it focuses on important mechanistic aspects using a variety of techniques while the NMR data provides an atomic-resolution map of key interactions between RBM39 and binding partners, which is very valuable. Hence, this manuscript should be acceptable for publication in Nature Communications, if authors can satisfy the minor comments below.

1. The 'applied' implications of this research on cancer therapeutics are important, however the nature of the data is actually mechanistic and more basic than applied. Hence, the Abstract excessively focuses on the existing drugs targeting RBM39 in cancers where this factor acts as an oncogene, and in the dependence of these drugs on DCAF15. I find this excessive because nowhere on the paper was this drug tested, and none of the experiments involved DCAF15. I recommend an extensive rewriting of the Abstract to focus on the main and very valuable mechanistic insights of this work, rather on the implications for drug design. This work is great on its own, without having to invoke too much the drug aspects.

We partly agree with the reviewer and adapted the abstract to focus more directly on our findings.

The discovery of the mode of action of aryl sulfonamides was a major achievement of the last 5 years in the field of acute myeloid leukemia research. This discovery enables a rational understanding of the mode of action of these anti-AML drugs, but the clinical output was not as good as expected. In light of the disappointing clinical evaluation, a problem has been identified: the aryl sulfonamides require high expression level of DCAF15 to actively induce the targeted degradation of RBM39 in cancer cells.

By investigating the molecular mechanisms of RBM39 autoregulation, we bring mechanistic insights that could be used to deplete RBM39 from cancer cells independently of DCAF15. We are not interested in the existing drug but rather in discovering alternative therapeutic strategies triggering the depletion of RBM39 in cancer cells independently of DCAF15.

2. The transcriptomic data analysis is a bit limited, as more insights could be derived from the RBM39 depleted cells. Can we compare the RBM39 regulated events (changed in knockout and rescued by the full-length) to other transcriptomes, or to cancer samples? Just mine a bit more the existing data.

In this manuscript, we employed RNA-Seq to study the effect of RBM39 depletion on RNA metabolism in HeLa cells with the aim to identify alternative splicing events for subsequent functional studies. Since HeLa cells are genetically abnormal and unrelated to leukemia cancer cells, a deeper investigation of the RNA-Seq data is unlikely to yield relevant insight into the role of RBM39 in AML and comparisons with other datasets obtained using different cell types and methodologies are difficult in our experience.

More detailed transcriptomic analyses carried out in clinically relevant cell types were previously performed, for example by Wang, E. et al. (Targeting an RNA-Binding Protein Network in Acute Myeloid Leukemia. Cancer Cell 35, (2019)).

3. In particular, one is kept wondering whether the RBM39 poison exon is found more included in RBM39 overexpressing cancers. Does this exon trigger NMD in actual tumor samples, considering that NMD may be repressed in them? The TCGA data should shed light on this.

We thank the reviewer for this helpful idea. As suggested, we explored the alternative splicing of RBM39 pre-mRNA in RNA-Seq data from The Cancer Genome Atlas (TCGA) using TCGASpliceSeq. We found that the median inclusion efficiency of exon 2b was highest in AML compared to all other cancer subtypes in the TCGA (Fig 1g). Given that total RBM39 levels are also highest in AML, this result indicates that the autoregulation mechanism operates in human tumour samples. We added a plot showing the percentage of poison exon inclusion for the different cancer types in the TCGA dataset in Fig. 1 panel g:

4. While this work focuses on the RRM, authors wasted a great opportunity to test the N-terminal RS domain. Can the RS-domain deficient mutants rescue the RBM39 functions?

We agree with the reviewer and carried out experiments to study the role of the RS domain in RBM39-dependent splicing.

Since the RS domain of RBM39 contains the nuclear localization signal, we designed a FLAG-tagged mutant of RBM39 lacking the RS domain fused to an artificial SV40 NLS.

We found that the RS domain is essential for RBM39 splicing and controls the localization of RBM39 to nuclear speckles. In addition, the RS domain is required for the interaction between RBM39 and U2 snRNP.

The results are summarized in a novel figure 3:

5. At the end of page 4, authors mention that the rescue experiment ensures that the splicing targets elucidated by knockdown are direct. While this rescue is key to rule out off-target effects of the knockdown, it is still possible that the rescue RBM39 regulates splicing factor(s) that in turn regulate the splicing event. Instead, the direct splicing targets are supported by experiments showing direct binding of protein to the RNA in cells, and/or by in vitro splicing/binding experiments with purified components. Please rephrase.

The sentence:

“Among the 1000 most significant events, 73.3% of the transcriptional changes, 82.7% of the exon skipping events and 93.8% of the intron retention events were rescued, indicating that the alterations are directly caused by the activity of RBM39.”

was rephrased to

“Among the 1000 most significant events, 73.3% of the transcriptional changes, 82.7% of the exon skipping events and 93.8% of the intron retention events were rescued, indicating that the alterations are not caused by off-target effects of the RBM39 siRNAs.”

6. Figures 1F and 2E, these RT-PCR experiments should be done in experimental triplicates with means and standard deviations like in Figure 7B.

These experiments have been performed in triplicates. Since they are redundant with Figure 2g, a single quantification of these experiments is shown in Fig. 2g.

7. Figure 7B can be a bit misleading, because the mutant minigenes are not expected to affect the endogenous RBM39 splicing pattern, make sure this is clear.

In Figure 7B (now Extended data Fig. 10C), we also probed the splicing of the endogenous RBM39 as a control to confirm that RBM39 is functionally impaired in the knockdown conditions.

Reviewer #3 (Remarks to the Author):

In this manuscript, "The cancer-associated RBM39 bridges the pre-mRNA, U1 and U2 snRNPs to regulate alternative splicing", Campagne et al. used functional and structural approaches to understand the role of RBM39 in RNA metabolism. They demonstrated that RBM39 is actively participates in splice site selection and autoregulates through a negative feedback loop mechanism by controlling the inclusion of the poison exon in its own pre-mRNA. All three RNA recognition motifs (RRM1/2/3) are essential for contacting the splicing machinery at both splice sites as well as its pre-mRNA in order to achieve the autoregulation. The authors provide important data to manipulate this negative feedback loop mechanism in order to trigger the depletion of RBM39 independently of DCAF15.

Thus, this manuscript might provide new therapeutic strategies for patients with leukemias and lymphomas that express high levels of DCAF15. However, this manuscript needs to be improved as follows:

Major concern:

In the results section describing Figure 7, the authors write, "Compared to the endogenous RBM39 mRNA, poison exon inclusion was more efficient in the minigene, likely because the minigene mRNA is less susceptible to nonsense-mediated decay". First please add a reference to support this claim. If indeed the HBB minigene is sensitive to the NMD pathway you need to add an additional mini-gene that is not sensitive to NMD, and show how RBM39 influences the inclusion level of the poison exon in this mini-gene. As the HBB minigene does not mimic the genomic environment of the RBM39 gene, and the minigene is not under the regulatory system of the endogenous promoter, all results obtained from this minigene need to be strengthened. For example, as elongation rate influences inclusion of exons, promoters can modulate exon inclusion. Therefore, it is important to show the interplay between RBM39 and the inclusion level of the poised exon using two different promoters (a strong and a weak one). Further, RNA secondary structure can be influenced by the elongating polymerase. To confirm that the inclusion of poison exon is independent of RBM39 and controlled by an inhibitory stem loop, the authors should choose another two minigenes and insert the poison exon and approximately 100 nucleotides of its flanking intronic region into these minigenes and evaluate splicing.

This issue was also raised by reviewer #1, showing that there is a general problem regarding the clarity of the finding here. As mentioned above, we repeated the experiments using an RBM39 minigene

which provides more endogenous context and replicates the regulation of poison exon inclusion more accurately. We consider this approach of providing more endogenous context more relevant than inserting the poison exon into additional heterologous minigenes. The conclusions are identical to those drawn from the old data and are summarized in Figure 8 while the splicing assays performed in the context of the β -globin have been moved to Extended data Fig. 10. Here are the new results obtained in the context of the RBM39 minigene:

We agree that exon inclusion level might be dependent on transcription rate. However, given that all our minigenes have the same promoter, we conclude that the differences between them are promoter-independent.

Minor concerns:

The authors write, “Furthermore, the RS domains of RBM39 and U2AF2 were shown to promote liquid–liquid phase separation and to favor 3′ splice site recognition”. It is unclear how this sentence is relevant to the rest of the manuscript.

This sentence has been removed according to the reviewer comment.

Figure 1C: The functional contribution of each RRM to the three intron retention events detected by DEXseq. Schematics of these genes and the exact intron retention events that were examined should be shown so that it is clear what events the authors are referring to.

The Extended data Fig. 2 has been clarified according to the reviewer suggestion.

Figure 2C: “All the constructs were moderately expressed and none of the deletions affected RBM39 subcellular localization”. Please add a graph showing a quantification to support this sentence.

A graph showing a quantification was added in Extended data Fig. 2, according to the reviewer suggestion.

Extended figure 2A: It is unclear which intron retention events were used for the validation of the RNA-seq data shown for TPP1, MBD1, and PAPOLA. Give a number for each intron or exon and relate it to UCSC genome browser.

Exon numbers and chromosomal location of each intron are now clearly indicated in Extended data Fig. 2.

Figure 3A: Figure 7d illustrates a model for poison exon inclusion within the context of RBM39 autoregulation, showing that the RBM39 domain interacts with U1 (via RRM1) and with U2 and U2AF65 (via RRM3) and that RBM39 interacts with U1, U2, and U2AF65 form a complex that is crucial for exon 2b inclusion level. Therefore, please add western blotting showing that U2AF65 co-precipitates with all snRNPs.

Since U2AF65 is a well-established constituent of spliceosomal complexes E, A, B and C (see spliceosome database: <http://spliceosomedb.ucsc.edu/>), we expect U2AF65 to co-purify with all snRNPs even if its recruitment to certain 3′-ss requires the help of splicing factors. We therefore doubt that such an experiment would help us understand the mechanism of RBM39 poison exon inclusion.

However, as described above, we have now adapted our model regarding the interaction between RBM39 and the U1 snRNP based on further IP and *in vitro* RNA-binding experiments.

REVIEWERS' COMMENTS

Reviewer #1 (Remarks to the Author):

The authors have addressed the previous concerns and especially corrected some statements and conclusions that were not supported by their data. Some additional experiments were performed to clarify some aspects. Although there is still some things to clarify in the future I recommend publication of the revised manuscript.

Reviewer #2 (Remarks to the Author):

All ok now, the revised manuscript is acceptable.

Reviewer #3 (Remarks to the Author):

After reading the revised manuscript again titled "Molecular basis of RNA-binding and autoregulation by the cancer-associated splicing factor RBM39," I would like to express my support for its publication in Nature Communications. The revised manuscript provides clear insights into the molecular mechanisms underlying the RNA-binding and autoregulation of the cancer-associated splicing factor RBM39. In light of these improvements, I believe that the manuscript is now ready for publication in Nature Communications. I would like to extend my congratulations to the authors for their excellent work, and I wish them continued success in their research endeavors.